

# Characterization of a catalyst-based total nitrogen and carbon conversion technique to calibrate particle mass measurement instrumentation

Chelsea E. Stockwell[1,2], Agnieszka Kupc[1,2], Bartlomiej Witkowski[1,2,3], Ranajit K. Talukdar[1,2], Yong Liu[4], Vanessa Selimovic[5], Kyle J. Zarzana[1,2], Kanako Sekimoto[1,2], Carsten Warneke[1,2], Rebecca A. Washenfelder[1], Robert J. Yokelson[5], Ann M. Middlebrook[1], James M. Roberts[1]

[1]NOAA Earth System Research Laboratory (ESRL), Chemical Sciences Division, Boulder, CO 80305, USA

[2]Cooperative Institute for Research in Environmental Sciences, University of Colorado, Boulder, CO 80309, USA

[3]University of Warsaw, Faculty of Chemistry, al. Żwirki i Wigury 101, 02-089, Warsaw, Poland

[4]University of Colorado Denver, Department of Chemistry, Denver, CO 80217, USA

[5]University of Montana, Department of Chemistry, Missoula, MT 59812, USA

*Correspondence to*: C. E. Stockwell (Chelsea.Stockwell@noaa.gov); J. M. Roberts (James.M.Roberts@noaa.gov)

**Abstract.** The chemical composition of aerosol particles is a key aspect in determining their impact on the environment. For example, nitrogen (N)-containing particles impact atmospheric chemistry, air quality, and ecological N-deposition. Instruments that measure total reactive nitrogen ($N_r$ = all nitrogen compounds except for $N_2$ and $N_2O$) focus on gas-phase nitrogen and very few studies directly discuss the instrument capacity to measure the mass of $N_r$ –containing particles. Here, we investigate the mass quantification of particle-bound nitrogen using a custom $N_r$ system that involves total conversion to nitric oxide (NO) across platinum and molybdenum catalysts followed by NO-$O_3$ chemiluminescence detection. We evaluate the particle conversion of the $N_r$ instrument by comparing to mass derived concentrations of size-selected and counted ammonium sulfate (($NH_4)_2SO_4$), ammonium nitrate ($NH_4NO_3$), ammonium chloride ($NH_4Cl$), sodium nitrate ($NaNO_3$), and ammonium oxalate (($NH_4)_2C_2O_4$) particles determined using instruments that measure particle number and size. These measurements demonstrate $N_r$-particle conversion across the $N_r$ catalysts that is independent of particle size with 98 ± 10% efficiency for 100 − 600 nm particle diameters. We also show conversion of particle-phase organic carbon species to $CO_2$ across the instrument's platinum catalyst followed by a non-dispersive infrared (NDIR) $CO_2$ detector. We show the $N_r$ system is an accurate particle mass measurement method and demonstrate its ability to calibrate particle mass measurement instrumentation using single component, laboratory generated, $N_r$-containing particles below 2.5 µm in size. In addition we show agreement with mass measurements of an independently calibrated on-line particle-into-liquid sampler directly coupled to the electrospray ionization source of a quadrupole mass spectrometer (PILS-ESI/MS) sampling in the negative ion mode. We obtain excellent correlations ($R^2$ = 0.99) of particle mass measured as $N_r$ with PILS-ESI/MS measurements converted to the corresponding particle anion mass (e.g. nitrate, sulfate, and chloride). The $N_r$ and PILS-ESI/MS are shown to agree to within ~6% for particle mass loadings up to120 µg m$^{-3}$. Consideration of all the sources of error in the PILS-ESI/MS technique yields an overall uncertainty of ± 20% for these single component particle streams. These





results demonstrate the $N_r$ system is a reliable direct particle mass measurement technique that differs from other
particle instrument calibration techniques that rely on knowledge of particle size, shape, density, and refractive index.
**1 Introduction**
Aerosol particles are a key component of the atmospheric chemical environment as they have climate, human
health, and ecosystem effects (Pöschl, 2005; IPCC, 2013). Measuring aerosol particle chemical composition is a
challenging endeavor that has been the subject of a great deal of innovation in the past few decades (Jayne et al., 2000;
Weber et al., 2001). The calibration of these instruments has evolved to better detect speciated composition. Still,
there is a need for fundamental mass-based calibration techniques to place aerosol particle measurements firmly in
the context of other atmospheric chemical observations.
Nitrogen (N) compounds are major constituents of atmospheric aerosol and play a significant role in
atmospheric chemistry, radiative balance, air quality, and N-deposition in both terrestrial and aquatic ecosystems (Neff
et al., 2002; Liao et al., 2003; Forster et al., 2007; Cornell, 2010; Xu et al. 2012; Park et al., 2014; Fuzzi et al., 2015).
The relative contribution of N-compounds, specifically particulate nitrate, to total atmospheric particle mass is
expected to increase in the coming century due to a projected reduction in $SO_2$ and increasing $NH_3$ (Bauer et al., 2007;
Bellouin et al., 2011; Hauglustaine et al., 2014; Li et al., 2015), and already dominates in some urban and agricultural
environments (Haywood et al., 2008; Vieno et al., 2016). Excluding N-species in deposition studies contributes to
uncertainty in regional and global nitrogen budgets used to evaluate ecological, biogeochemical, and climate impacts
(Jickells et al., 2013; Cornell, 2010; Cape et al., 2011). Measuring individual N-species, classes of N-compounds, or
total N is challenging, and laboratory and field data are limited. For example, while there are a number of methods to
measure inorganic N species, particulate organic N is more difficult to quantify with fewer sampling and measurement
methods currently available for such a variety of compounds (Lin et al., 2010; Farmer et al., 2010; Lee et al., 2016).
Measuring the total N mass of atmospheric particles will improve our understanding of their role in nitrogen cycles
associated with sources such as agriculture or wildfires, and processes such as photochemical oxidation.
Several techniques exist to measure total reactive nitrogen ($N_r$), defined here as all atmospheric nitrogen
excluding $N_2$ and $N_2O$, which includes both gas (e.g. total odd nitrogen ($NO_y$), $NH_3$, amines, nitriles, nitrates, etc.)
and particle phase species (e.g. inorganic and organic N compounds). An established, rapid-response, robust technique
for measuring $N_r$ involves thermal and catalytic conversion to nitric oxide (NO) with detection by $O_3$
chemiluminescence. The catalyst material, temperature, and sampling methods dictate the efficiency, time resolution,
and speciation of measurements (Winer et al., 1974; Williams et al., 1998; Dunlea et al., 2007; Schwab et al., 2007;
Benedict et al., 2017). The chemiluminescence detection technique has been used to measure $NO_x$ ($NO + NO_2$; Parrish
and Fehsenfeld, 2000), total gas-phase $N_r$ (e.g. Hardy and Knarr, 1982; Horstman, 1982), individual reactive nitrogen
components (e.g. $NH_3$; Breitenbach and Shelef, 1973; Saylor et al., 2010), or subsets of nitrogen compounds by
removal of selected compounds using filters or denuders upstream (Prenni et al., 2014). Marx et al. (2012) completed
the only study to explicitly report quantitative conversion of particle-bound $N_r$ for a limited number of species,
however the results show a range of conversion efficiencies (78 – 142%). Several other studies assume at least some
(non-quantitative) particle conversion across their catalysts (Fahey et al., 1985, 1986; Prenni et al., 2014). To our



knowledge, no study selectively isolates particle-phase reactive nitrogen to assess the particle-phase contribution to total nitrogen signals from individual sources or in their atmospheric measurement. Here we characterize the particulate $N_r$ conversion in our converter consisting of heated platinum and molybdenum catalysts followed by rapid chemiluminescence detection using common inorganic atmospheric $N_r$-species including $(NH_4)_2SO_4$, $NH_4Cl$, $NaNO_3$, $NH_4NO_3$, and $(NH_4)_2C_2O_4$. The application of the converter coupled with $NO$-$O_3$ chemiluminescence, hereafter referred to as the $N_r$ system, to quantitatively convert and measure the sum of $N_r$ particle mass was evaluated using mass concentrations determined using traditional particle instrument calibration methods.

Organic carbon species are major constituents of aerosol particles (Jimenez et al., 2009) and are responsible for some of the more important climate and health impacts of particles (Pöschl, 2005). Calibration of measurement systems for organic carbon species is a challenging task since there are thousands of possible compounds of differing sizes, functional groups and therefore volatilities (Jimenez et al., 2016; Murphy, 2016a, b). A comprehensive, mass-based technique for organic aerosol species would be a highly-desirable addition to the current measurement technology. Theoretically, the high-temperature platinum catalyst in our system should convert carbon species to carbon-dioxide ($CO_2$) in the presence of air. Conversion of volatile organic compounds (VOCs) to $CO_2$ on high temperature precious metal catalysts is a well-developed technique (see for example the Pt catalyst used in Veres et al., 2010). Total organic carbon measurements using similar catalysts (e.g. palladium/alumina) followed by reduction to methane have been used previously (Roberts et al., 1998; Maris et al., 2003). By these methods, Roberts et al. (1998) confirmed efficient conversion of $C_1$-$C_7$ gas-phase compounds across the catalyst. Platinum-based catalysts are widely used and have been shown to be more efficient than palladium in oxidation studies (Schwartz et al., 1971; Kamal et al., 2016). Here we characterize the conversion efficiency of particle-phase organic carbon across our Pt catalyst by direct measurements using a LICOR non-dispersive infrared (NDIR) $CO_2$ analyzer. The current converter specifications coupled with both NO and $CO_2$ detectors allows simultaneous measurements of $N_r$ and total carbon ($C_y$).

Many traditional particle instrument calibration methods involve measurements of particle properties by inertial, gravitational, diffusional, electrical (e.g. sizing), thermal, or optical measurement devices (Chen et al., 2011). Generally, direct mass concentration calibration techniques involve off-line analysis of filters or semi-real time measurements (e.g. PILS combined with ion chromatography). More rapid techniques directly measure number concentrations and particle sizes. However, these methods often require knowledge of aerosol properties (e.g. composition, shape, density, refractive index) and sampling parameters (e.g. volumetric flow rate, pressure, temperature, relative humidity) in order to determine mass concentrations. While these instrument calibration techniques are well established for controlled laboratory generated aerosol standards, the $N_r$ system is an alternative that directly measures mass traced back to gas phase calibration standards instead of relying on particle size, shape, or refractive index.

In order to demonstrate the application of the $N_r$ system to directly measure particle mass to calibrate particle mass measurement instrumentation, we compare mass concentrations measured by a new approach of directly coupling a particle-into-liquid sampler to the electrospray ionization source of a quadrupole mass spectrometer (PILS-ESI/MS) for on-line mass analysis of water-soluble aerosols. The Particle-into-Liquid Sampler (PILS) is an





established technique developed to efficiently collect the water-soluble fraction of aerosol (Weber et al. 2001; Orsini
et al., 2003; Sorooshian et al., 2006). Here, we couple the PILS with an independently calibrated electrospray interface
followed by mass detection to obtain on-line mass measurements of single-component, laboratory generated, $N_r$-
containing aerosol that can be directly calibrated using the $N_r$ system.

5       In this work, we present the converter set-up, system methodology, and evaluate the particle-conversion

efficiency of a custom $N_r$ system for several atmospherically relevant $N_r$-containing particles. The conversion
efficiency of the $N_r$-catalyst was evaluated by comparing the $N_r$ mass signal with the mass calculated from instrument
calibration techniques that measure the particle number size distributions of laboratory-generated aerosols of known
composition. We then show the quantitative conversion of organic carbon across the instrument's platinum catalyst
followed by $CO_2$ detection. Finally we compare particle mass directly measured using a particle-into-liquid sampler
coupled directly to an electrospray ionization source and by the $N_r$ instrument. The primary objective of these
experiments is to characterize particle conversion in the $N_r$ system, and to investigate the capabilities of the $N_r$ system
as a calibration instrument that directly measures particle mass concentration.
**2 Experimental details**
**2.1 Description of the total reactive nitrogen ($N_r$) system**

16       Measurements of total reactive nitrogen, $N_r$, were accomplished by catalytic conversion to NO and detection

of the NO using a chemiluminescence instrument. This $NO-O_3$ chemiluminescence instrument is a custom-built
version of the common atmospheric monitoring instrument (Williams et al., 1998) and is calibrated directly with gas
phase standards of NO. All the $N_r$ species were converted to NO or $NO_2$ on a high temperature catalyst, and the $NO_2$
subsequently converted to NO on a lower temperature catalyst. The high-temperature catalyst system consisted of a
quartz tube (13 mm OD x 11 mm ID x 35 cm L) packed with 36 platinum (Pt) screens (Shimadzu Part No. 630-00105)
run at high temperature (750°C), shown in Fig. 1. The catalyst bed was confined to an 8 cm long section by dimples
in the quartz tube, and that section was positioned so that the gas reaching it had been equilibrated to 750°C, as
confirmed by a thermocouple probe. The flow through the catalyst was set to 1 standard L min$^{-1}$ via a downstream
flow controller. The Pt surface area was 126 cm$^2$ and the residence time was 0.1 s at 83.3 kPa and 750°C. Platinum
catalysts of this kind are also known to oxidize NO to $NO_2$, which has been the source of problems with some previous
systems that were designed to measure atmospheric ammonia ($NH_3$) (Schwab et al., 2007). In our system, the Pt
catalyst is followed by a molybdenum oxide (MoOx) catalyst consisting of a solid molybdenum tube (4.2 mm ID x
32 cm L) operated at 450°C, to which an 8 standard cm$^3$ min$^{-1}$ flow of pure hydrogen was added to create a stable
molybdenum oxide surface. Run in this manner, the MoOx surface did not require periodic treatment at higher
temperatures under reducing conditions as described by Williams et al. (1998). The NO chemiluminescence detection
scheme used for laboratory calibrations had a fundamental sensitivity between 6 and 7 counts per parts per trillion
(pptv) and the detection limit determined by the background signal in zero air was typically 0.15 pptv (4 σ) for a 1 s
measurement. The operation of this instrument during these experiments often required considerable de-tuning to keep





the instrument count rates below the roll-over point of the photon counting electronics (approximately 5 MHz), thus
the detection limit was closer to 0.1 ppbv for these measurements.
**2.1.1 Nitrogen-containing particles**
The measurement of particle phase $N_r$ requires decomposition or volatilization of the solid material, followed
by catalytic conversion to NO (or $NO_2$). Broadly, there are three types of $N_r$-containing particles, with a range of
thermal stabilities from volatile to refractory. First, there is considerable literature that indicates that small particles
composed of two major semi-volatile species, ammonium nitrate ($NH_4NO_3$) and ammonium chloride ($NH_4Cl$), will
dissociate to constituents $NH_3$ and $HNO_3$ (and HCl), when modestly heated to temperatures < 100°C (Huffman et al.,
2009; Hu et al., 2011). These materials will be readily converted on high temperature catalysts (e.g. platinum, Pt) as
gas phase $NH_3$ and $HNO_3$. The second type of $N_r$-containing particles include intermediate stability compounds
consisting mostly of nitro-organics (R-$NO_2$), organic nitrates (RO$NO_2$), and amine and ammonium salts of acids.
These compounds begin to decompose at relatively low temperatures. For example, thermal decomposition studies of
bulk ammonium oxalate (($NH_4$)$_2C_2O_4$) indicate that it begins to decompose at temperatures slightly above 200°C
(Usherenko et al., 1998). Similarly, bulk samples of ammonium sulfate (($NH_4$)$_2SO_4$) and ammonium bisulfate
(($NH_4$)$HSO_4$) decompose at approximately 150–250°C depending on water content (Kiyoura and Urano, 1970). Given
sufficient residence time, intermediate volatility compounds will start to convert to gas-phase products in the hot inlet
tubing and fully convert to NO (or $NO_2$) on a hot Pt surface (750°C). The third type of $N_r$-containing particles are
composed of refractory salts such as sodium nitrate ($NaNO_3$), which will be the most resistant to decomposition and
require contact with high temperature surfaces of the Pt catalyst. Studies of the thermal decomposition of $NaNO_3$ on
Pt surfaces indicate that NO is evolved starting at about 500°C. In summary, the existing literature suggests that the
thermal decomposition/conversion of $N_r$-containing particles to NO ($NO_2$) is thermodynamically feasible provided
there is sufficient residence time and surface area in the catalyst zone.
**2.2 Description of the PILS-ESI/MS**
A schematic of the PILS-ESI/MS is shown in Fig. 2. The Particle-into-Liquid Sampler (PILS; Brechtel
Manufacturing Inc., Hayward, CA) was developed by Weber et al. (2001) and collects water-soluble aerosol
compounds by growing particles into liquid droplets in a supersaturated water environment and then collecting the
droplets. A detailed description of the PILS used in these studies can be found in Sorooshian et al. (2006). The PILS
is an established water-soluble aerosol collection technique that has been coupled with various mass analysis methods
and was used previously by other laboratories in instrument evaluation studies (e.g. Drewnick et al., 2003; Takegawa
et al., 2005; Canagaratna et al., 2007).
The PILS sample flow was set to 100 µL min$^{-1}$ and was continuously mixed with an acetonitrile flow (100
µL min$^{-1}$). The 1:1 volume mixture of acetonitrile and water was directed toward the custom electrospray ionization
source (at ~10 µL min$^{-1}$) of a commercial quadrupole mass spectrometer (Balzers Instruments, QMG 422)operated in
negative ion mode for on-line analysis of selected water-soluble organic and inorganic compounds. The electrospray
interface involved sample injection at ambient pressure through a fused silica capillary tip (30 µM ID) with a 2.5 L



min$^{-1}$ N$_2$ sheath flow at a spray voltage of -3.5 kV. The MS instrument was modified from the negative-ion proton-
transfer chemical-ionization mass spectrometer (NI-PT-CIMS) described in Veres et al. (2008). The flow tube was
replaced with a stainless steel capillary inlet connected to the front region (I; shown in Fig. 2) held at ~300 Pa. Ions
were focused across this region using a planar DC ion carpet (Anthony et al., 2014) mounted in front of the orifice
leading to the second region (II). The ions were then accelerated through the collisional dissociation chamber (CDC)
and collimated in the octopole ion guide at a total pressure of ~1 Pa (region II). The ions were transferred to the
quadrupole mass spectrometer (region III). The electron multiplier detector was maintained at a pressure of less than
$6.6 \times 10^{-3}$ Pa.
The ESI/MS was calibrated using volumetrically and gravimetrically prepared liquid-phase standards of the
anions associated with the target compounds (e.g. SO$_4^{2-}$, NO$_3^-$, Cl$^-$) (Sigma Aldrich, St. Louis, MO). Anion-specific
calibration factors were calculated from linear least-squared fits of multi-point calibration curves. The uncertainty in
the slope resulted in a maximum uncertainty of ~10% for the compounds tested. The ESI flow rate, solvent
composition, analyte chemical properties, and matrix effects potentially impact the ionization and transmission
efficiencies of compounds (Kostiainen and Kauppila, 2009). For these reasons, experiments were performed under
similar, or as close to identical, conditions as the calibrations for instrument evaluation. For purposes of this
comparison, matrix effects were assumed to be negligible for tests sampling single-component aerosols. The limits of
detection for the anions measured with the PILS-ESI/MS were below ~0.1 µg m$^{-3}$ for the current system and sampling
conditions. Sorooshian et al. (2006) discuss volatility losses in the PILS for several inorganic species and reported
negligible loss with a collection efficiency of ≥96% for mass loadings of Cl$^-$, SO$_4^{2-}$, and NO$_3^-$ ranging from 1-140 µg
m$^{-3}$. Additionally, Orsini et al. (2003) showed the collection efficiency of ≥95% for particles as small as 30 nm
diameter for a 15 L min$^{-1}$ sample flow rate. Ammonium (NH$_4^+$) is the major ion susceptible to volatilization as shown
in Ma (2004), who indicated an underestimation of ~15%. In this study, because we were operating in the negative-
ion mode, we did not measure NH$_4^+$ directly.
**2.3 Particle generation, measurement, and characterization**
Several aerosols were generated including polystyrene latex spheres (PSL; Nanosphere size standards,
Thermo Fisher Scientific Inc., Waltham, MA), NH$_4$NO$_3$, (NH$_4$)$_2$SO$_4$, (NH$_4$)$_2$C$_2$O$_4$ (Sigma Aldrich, St. Louis, MO),
NH$_4$Cl (J.T. Baker Chemical Co., Phillipsburg, NJ), and NaNO$_3$ (Fisher Scientific, Hampton, NH). Aerosol particles
were generated by atomization of aqueous solutions of pure compounds in distilled water (~0.5– 6 g L$^{-1}$) using a
custom-built Collison-type atomizer (Liu and Lee, 1975) in a dry particle-free nitrogen flow. The output flow was
dried using a silica gel diffusion dryer to a relative humidity less than 10%. The dry polydisperse particles were then
size-selected using a custom-built differential mobility analyzer (DMA; Knutson and Whitby, 1975). The DMA was
operated at a sample flow of 0.3–0.5 volumetric L min$^{-1}$ and a ratio of 10:1 between the sheath and sample flow. The
monodisperse particles were diluted with ultra-high purity filtered zero air (range 1–10 L min$^{-1}$) before entering a
mixing vessel. In instances where a mixing vessel was not available, a segment of smaller diameter tubing was added
in-line to promoted mixing prior to the flow being divided among the instruments. A condensation particle counter
(CPC; 3022A, TSI Inc., Shoreview, MN) (Stolzenburg and McMurry, 1991) continuously measured the particle



number concentration of the output flow following dilution. We measured the flow pre- and post-sampling using a low-flow DryCal (Mesa Laboratories, Lakewood, CO) and estimate an uncertainty in the CPC flow rate calibration to be ± 1%. During several experiments, the aerosol flow was split and sampled by an ultra-high sensitivity aerosol spectrometer (UHSAS; Droplet Measurement Technologies, Longmont, CO) to continuously measure the particle concentration and size distribution for particles with diameters between ~63 and 1000 nm.

In these experiments particle diameters from 100 to 600 nm were selected and the multiply-charged particles in the size distribution were accounted for as described below. For the liquid concentrations and atomizer conditions we used, the DMA output size distribution is a multi-peaked population consisting not only of singly charged particles but also particles with multiple (mostly two or three) charges. The multiply charged particles can contribute significantly to the overall mass and must be considered when calculating particle mass. The distributions of singly, doubly, and triply charged particles can vary depending on the solution concentration. We measured atomized size distributions using the scanning mobility particle sizer (SMPS; Wang and Flagan, 1990) function of the DMA (physical diameter, $D_p$ = 1-1000 nm). The DMA transfer theory (Knutson and Whitby, 1975; Stolzenburg, 1988) with Wiedensohler's (1988) steady-state charge distribution approximation was used to estimate the fraction of multiply charged particles contributing to the CPC number concentration for each diameter setting. There are a number of possible sources of uncertainty using these methods that may include particle losses, DMA transfer function uncertainty, counting uncertainty, and inversion errors. Consequently, the size distribution of particles selected at a particular voltage and flow setting of the DMA was examined using the UHSAS. UHSAS particle sizing is a function of the amount of light scattered onto the photodetectors. The quantity of scattered light, however, depends not only on the particle size, but also on the composition-dependent particle refractive index (Bohren and Huffman, 1983; Liu and Daum, 2000; Hand and Kreidenweis 2002; Rosenberg et al., 2012). The UHSAS manufacturer recommended calibration uses PSL microspheres, which are well characterized and have known refractive index (n = 1.58) and shape. Because the UHSAS sizing is sensitive to particle refractive index, a new sizing calibration curve was produced for each studied particle type (i.e. refractive index) (Kupc et al., 2017). Considering this, we used the DMA, with sizing accuracy ~ ± 2.5% and NIST-traceable PSLs for 150 –500 nm spheres as our calibration standard. The UHSAS sizing was recalibrated by using the DMA to select particles of known size for each of the aerosol types studied. A different UHSAS calibration curve was produced and used for each aerosol type (e.g. Kupc et al., 2017). These calibration curves were used to retrieve accurate particle size distributions so that the multiply charged particles were properly accounted for.

Previous laboratory studies show UHSAS and CPC number concentration comparisons in excellent agreement (Cai et al., 2008; Kupc et al., 2017), however, occasionally only a ~90% counting efficiency for the UHSAS was observed when compared to the CPC. These differences are attributed to particle coincidence at high concentrations (> 1000 cm$^{-3}$), and to inefficient particle mixing before reaching the instruments. Corrections for particle coincidence were applied (Kupc et al., 2017) though we expect differences due to particle mixing adds an additional 10% uncertainty to the measurements. For these reasons, we used the UHSAS size distributions to estimate the fraction of singly, doubly, and triply charged particles together with the total particle number taken from the CPC measurement to exclusive particle mass from total volume and density. The UHSAS and CPC measured particle



number concentrations were generally within 10% of each other, however, the CPC values did not require coincidence
corrections and had a better signal to noise ratio.
**3 Results and discussion**
**3.1 Characterization of the $N_r$ system**
**3.1.1 $N_r$ gas-phase conversion efficiency**

6       We verified the efficiency of conversion of a range of gas phase $N_r$ compounds in this catalyst system using

calibrated gas mixtures or standard streams and auxiliary analysis methods. We compared the total $N_r$ signal measured
as NO, where NO was calibrated using NO standards in nitrogen (Scott-Marrin Inc., Riverside, CA) to the known
amount specified by the calibration method. The conversion efficiencies are summarized in Table 1 and range from
95% to 110%. The values were based on the ratios of the $N_r$ measured as NO to the expected values specified by each
calibration method. The uncertainties in the measured conversion efficiencies encompass the propagated errors in each
calibration method. For example, the largest uncertainty in the $NH_3$ conversion efficiency was the $NH_3$ UV absorption
cross section at 184.9 nm (value of $4.4 \pm 0.3 \times 10^{-18}$ cm$^2$ taken from Neuman et al., 2003). It is possible that there were
$N_r$ compounds in the standard stream aside from $NH_3$ that were responsible for the result being >100%. However, the
fact that the determination was above 100% for both a permeation source and a gas-phase mixture (3.1 ppmv in $N_2$)
implies that the UV absorption cross section is high by 5-10% or that there were contaminants in both calibration
sources. $NH_3$ is one of the more important reactive nitrogen species in the atmosphere-biosphere system and is
thermodynamically one of the more difficult to convert. Compounds considered $NO_y$ species, such as nitric acid,
acetyl peroxynitrates, and alkyl nitrates were not studied in this work (aside from $NO_2$), since they are known to be
converted at high efficiency on precious metal (Fahey et al., 1986) or molybdenum oxide (Winer et al., 1974) catalysts.
The resulting uncertainties in the $N_r$ measurement are estimated to be $\pm$ 10% based on comparisons of measured NO
signals to individual $N_r$ compound calibrations.

23       The conversion of nitrous oxide ($N_2O$) is a potential interference in the $N_r$ method as $N_2O$ is not typically

considered a reactive nitrogen compound in the troposphere. Several experiments were conducted to determine the
extent of this potential interference using a 10.1 ppmv $N_2O$ standard. The resulting conversion efficiency ranged from
0.03% to 0.05% in dry and humidified air respectively. These can be considered upper limits for this interference as
we cannot be completely sure that there were no $N_r$ contaminants (e.g. $NO_2$) in the $N_2O$ standard. This conversion
efficiency upper limit is a negligible interference in the $N_r$ measurements in ambient air or zero air matrices, and
likewise will not be significant in biomass burning sources given that $N_2O$ enhancements in fresh biomass smoke are
generally not observed or contribute minimally to total nitrogen (Griffith et al., 1991). $O_3$ is another potential source
of gas-phase interference due to the decomposition of $O_3$ to $O_2$ + O, followed by reaction of O with $N_2O$ at high
temperature to form NO. However, the NO production in the O + $N_2O$ reaction is an approximately 20% channel with
a net rate constant of approximately $1 \times 10^{-15}$ cm$^3$ molecule$^{-1}$ s$^{-1}$ at 750°C (NIST 2017). If all the O atoms from 70
ppbv of $O_3$ were available for reaction with an ambient level of $N_2O$ (340 ppbv), then the 0.1 sec residence time in the



convertor would result in approximately 28 pptv of NO, an upper limit that is clearly a negligible amount in almost
any atmospheric context.

### 3.1.2 $N_r$ system set-up and response

The atomizer output was diluted with particle-free nitrogen and ultra-pure zero air, therefore, the $N_r$
measurement should theoretically be attributed to particles only since no detectable gas-phase nitrogen is added to the
sample stream. However, equilibration within the sample lines may result in outgassing and formation of gas-phase
compounds affecting total $N_r$ detection. Fig. 3(a) shows the initial response of the $N_r$ system in cleaned inlets for
NaNO$_3$. The $N_r$ mass signal tracks the CPC-derived aerosol mass features closely as the aerosol source concentrations
fluctuate. Additionally, as different particle sizes are selected by the DMA for $(NH_4)_2SO_4$ (Fig. 3(b)), changes in the
total $N_r$ response is fast and precisely tracks the changes in the CPC signal. The potential gas-phase constituents
equilibrating in the lines from aerosols in this study include $HNO_3$, HCl, and $NH_3$. If these compounds formed before
reaching the $N_r$ catalyst it is likely adsorption and desorption from inlets and tubing surfaces would occur (e.g. Neuman
et al., 1999; Yokelson et al., 2003). As an example, the presence of $NH_3$ in Fig. 3(b) (or $HNO_3$ in nitrate containing
particles) would be indicated by a delayed and lengthened rise/fall in the $N_r$ response with sudden changes to the input
concentrations. However, the total $N_r$ response precisely tracks the CPC signal suggesting that gas-phase $NH_3$ was not
present in significant quantities. In experiments at exceptionally high aerosol loading of $(NH_4)_2C_2O_4$ (up to several
ppmv of total $N_r$, i.e., several thousand µg m$^{-3}$) $N_r$ signal "tailing" was observed suggesting that $NH_3$ was scavenging
to the walls of the inlet before the heated quartz tubing.
Marx et al. (2012) reported calculated conversion efficiencies in air sampled from a small chamber for
NaNO$_3$, $NH_4NO_3$, and $(NH_4)_2SO_4$ to be 78, 142, and 91%, respectively. The authors suggested the overestimation of
$NH_4NO_3$ was a result of its semi-volatile properties under ambient conditions that led to the formation of gaseous $NH_3$
and $HNO_3$ in the chamber. For these reasons, we limit the background artifacts and volatilization effects that may have
occurred during chamber filling and sampling in Marx et al. (2012) by sampling immediately following solution
atomization through conductive tubing at relatively high sample flow rates. Additionally, we use a DMA to size-select
the atomized polydisperse aerosol to evaluate the particle conversion efficiency at several different diameters (100 −
600 nm in 50 nm increments) to investigate the volatilization effects and conversion efficiencies of smaller particles
for the extended list of $N_r$-containing aerosols studied in our work.

### 3.1.3 $N_r$-particle conversion efficiency

The voltage scanning (SMPS) function of the DMA and number concentration measurements by the CPC is
a conventional method to determine particle size distributions, and for calculating particle mass from total volume and
density, assuming spherical particles. For the total nitrogen measurements, the total particle-bound $N_r$ mixing ratios
were retrieved and converted to mass concentrations for each corresponding salt. Figures 4(a-d) show the calculated
vs measured mass concentrations (µg m$^{-3}$) for particles of different composition and diameter. The plots show that a
strong correlation ($R^2 > 0.98$) and good agreement was obtained for smaller particles (50 − 200 nm) with slopes ranging





from 0.86 – 0.97, while for larger particles (≥250 nm) the mass calculated values were sometimes as much as >50%
too high. The $R^2$ for all particles including ≥ 250 nm ranged from 0.71 – 0.85 with slopes of 1.08 – 1.36.
For larger particles, we used a UHSAS to determine the size distribution of multiply-charged species exiting
the DMA. The SMPS inversion-derived size distributions were generally broader than the UHSAS size distributions,
though agreement improved at increased scan times. Small differences in the size distribution recovered from the
voltage scans at larger diameters (> 200 nm) affected the mass distribution considerably because particle mass scales
with diameter cubed. A possible explanation is that we are not correctly accounting for the delay time from the DMA
exit to the CPC, therefore the particle counts did not correspond to the correct size designated from voltage scanning
and this likely skewed the size distribution relative to the true distribution (Collins et al., 2002). Methods for limiting
these effects exist (Russell et al., 1995; Collins et al., 2002) including slower voltage scan rates. However, our results
demonstrate the added challenges in particle mass determination using estimated size distributions from the SMPS
method. Other aerosol measurement techniques (e.g. the Particle Time of Flight mode of the Aerosol Mass
Spectrometer; DeCarlo et al., 2006) directly measure size distributions or instead measure polydisperse aerosol and
the instrument and inversion-algorithm corrections required using the SMPS are avoided. Therefore, we instead
measure the size distributions directly using the UHSAS with particle concentration measurements (by either the CPC
or UHSAS) to evaluate the $N_r$ particle conversion in the $N_r$ system.
For the aerosol mass concentrations ($\mu g\ m^{-3}$) calculated using UHSAS particle size distributions, we refer to
these values as UHSAS calculated mass. Comparisons of the mass directly measured as $N_r$ versus UHSAS calculated
mass concentrations for atomized solutions of $NaNO_3$, $(NH_4)_2SO_4$, $NH_4Cl$, and $(NH_4)_2C_2O_4$ are shown in Fig. 5 with
orthogonal distance regression lines with slopes that range from 0.910 – 1.06 for concentrations from ~0–70 $\mu g\ m^{-3}$.
The instruments are highly correlated ($R^2 = 0.90 – 0.99$) and the fits indicate that for the salts tested there is quantitative
conversion of particulate nitrogen, to within the combined uncertainties of the methods, independent of diameter
(range: 100 – 600 nm). More detailed particle conversion efficiencies by size are shown in Table 2 for each aerosol
tested. On average across all size ranges the results indicate 97 ± 7%, 101 ± 5%; 100 ±10%, and 93 ± 5 % particle
conversion efficiencies for $NaNO_3$, $(NH_4)_2SO_4$, $NH_4Cl$, and $(NH_4)_2C_2O_4$, respectively. The largest deviation from the
one-to-one line occurred for $(NH_4)_2C_2O_4$, which may imply some ammonia loss, though the agreement is generally
still within 10% for most particle sizes.
For the case of $NH_4NO_3$, the UHSAS measured size distribution peaked at significantly lower diameters than
expected based on the DMA size selection. This difference has been reported previously (Cai et al., 2008; Womack et
al., 2017), though to a lesser extent (~8%) than observed here (up to 30%). Possible explanations for these differences
could include vaporization/evaporation effects, residual water in the particles, surface effects, or differences in
electrical mobility diameter and geometric diameter due to non-sphericity as discussed in DeCarlo et al. (2004). For
these reasons, we made no attempt to characterize $NH_4NO_3$ behavior in either the DMA or UHSAS and refer to Sect.
3.2 for mass concentration comparisons of polydisperse aerosol measured using separate mass measurement
techniques (both the $N_r$ system and PILS-ESI/MS). It is worth noting that $NH_4NO_3$ is one of the more volatile
compounds included in this study and it is reasonable to expect similar particle conversion efficiencies in the $N_r$ system
catalysts for $NH_4NO_3$ as the other species tested (Table 2).





### 3.1.4 $N_r$ measurements of biomass burning

As an example of both gas and particle measurements using the $N_r$ system, we follow with a brief discussion of N emissions from biomass burning. The primary gaseous N-compounds in biomass burning plumes include NO, $NO_2$, $N_2$, $NH_3$ and to a lesser extent HCN, $CH_3CN$, HONO, HNCO (Lobert et al., 1990; Lobert et al., 1991; Kuhlbusch et al., 1991; McMeeking et al., 2009; Burling et al., 2010; Stockwell et al., 2014; 2015) and other $N_r$-containing gases. Figure 6 shows results obtained from a representative fire (Fire 047) from the Fire Influence on Regional and Global environments Experiment (FIREX) 2016 Missoula Fire Lab study (https://www.esrl.noaa.gov/csd/projects/firex/). Figure 6(a) shows the co-measured $N_r$ and NO concentrations (ppmv). The majority of the $N_r$ system's response is due to the sum of gas-phase $N_r$-constituents that were measured by a Fourier transform infrared spectrometer (FTIR; Selimovic et al., 2017), an $H_3O^+$ chemical ionization mass spectrometer (Koss et al., 2017), and a broadband cavity enhanced extinction spectrometer (Min et al., 2016) (Fig. 6(b)). At the beginning of the burn (pre 10:23 AM) the average relative percent difference between the total nitrogen signal and the sum of individually measured gas-phase compounds is ~16%, which is less than the combined error of the individual measurements. There is greater disagreement shown in Fig. 6(c) (difference is up to ~1 ppmv; up to ~50% relative percent difference) during other stages of the fire. We have shown in our laboratory experiments that there is quantitative $N_r$ particle conversion across the $N_r$ catalyst, therefore, it is likely that particulate ammonium contributes to the excess $N_r$ signal measured during periods dominated by smoldering combustion, while particulate nitrate likely accounts for some $N_r$ signal during the flaming dominated stages as shown in Fig. 6. By confirming particulate $N_r$-conversion in this system, it is possible that a total N budget can be reconstructed for additional laboratory fires measured during the FIREX laboratory study where individual particle phase $N_r$ data are available.

### 3.1.5 Carbon conversion efficiency of Pt catalyst

The high-temperature platinum catalyst (Fig. 1) in the $N_r$ instrument should quantitatively convert carbon containing species to carbon-dioxide ($CO_2$) in the presence of air. Gas-phase carbon conversion across similar precious metals has been studied extensively (see for example the Pt catalyst used in Veres et al., 2010). Therefore, adding a $CO_2$ analyzer to the configuration allows for simultaneous measurements of $N_r$ and $C_y$.

For the following experiments, the total flow through the Pt catalyst was increased slightly (~1.5 sL min$^{-1}$) and was then split after the Pt and before the MoOx catalyst, with the smaller flow (0.5 sL min$^{-1}$) directed through the LICOR 6251 (LI-6251; Lincoln, NE) $CO_2$ analyzer, and the main flow directed through the MoOx catalyst and the NO-$O_3$ chemiluminescence detector. The LICOR instrument was internally referenced to scrubbed zero air. The conversion of compounds that contain both N and C atoms can then be measured simultaneously using the NO-$O_3$ chemiluminescence detector and LI-6251 detector in parallel. At ambient $CO_2$ levels, it is challenging to retrieve reliable measurements since the signal relative to the background abundance of $CO_2$ is small. The approach described here relies on using ultra-pure air for aerosol generation and carrier gas flow, therefore ambient air is eliminated. The LI-6251 was calibrated with sub-5 ppm $CO_2$ standards (Scott-Marin Inc., Riverside, CA) in ultra-pure air. Due to the





low signals levels and the uncertainty of the low concentration $CO_2$ standards, the overall accuracy of the $CO_2$
measurements present in this work up to 1 ppmv is ± 10% for 10 second averages.
The efficient conversion of gas-phase C-compounds in our catalyst system was confirmed using a CO
standard in air, and a combination $CO_2$, CO, $CH_4$ standard in air. The following discussion focuses on the conversion
of particle-phase organic compounds (OC). The efficient conversion of $N_r$-containing particles was demonstrated in
Sect. 3.1.3 for the range of N oxidation states. We are confident these results extend to other $N_r$-containing particles,
which is supported by the extensive list of $N_r$ gases efficiently converted as shown in Table 1. Therefore, we expect
that the resulting $N_r$ and $C_y$ signals from each detector will be in proportion by dividing the result by the number of
carbon and nitrogen atoms in the parent molecule to give the standard concentration on a molar basis. Polydisperse
particulate OC was generated from solution following an $N_2$ purge to eliminate carbonate from the solution. Aerosol
particles from solutions of anthranilic acid ($C_7H_7NO_2$, 2-aminobenzoic acid, Sigma Aldrich), threonine ($C_4H_9NO_3$, 2-
amino-3-hydroxybutanoic acid, Sigma Aldrich), tryptophan ($C_{11}H_{12}N_2O_2$, 2-amino-3-indolylpropanoic acid, Sigma
Aldrich), and quinine ($C_{20}H_{24}N_2O_2$, Sigma Aldrich) were tested. These compounds were chosen based on their water
solubility to avoid the use of organic solvents. An example of the $N_r$ and $C_y$ response is shown in Fig. 7 for threonine
(see Fig. S1 for additional compounds). The relative difference between the $N_r$ and $C_y$ measured concentrations (up
to several hundred ppbv) is less than 10%, which is within the propagated uncertainties of the $CO_2$ calibration
standards and both detection methods.
Initial tests with $(NH_4)_2C_2O_4$ proved more challenging as the low C number required large polydisperse
aerosol loadings (several ppmv) to be measured reliably by the LICOR. During these instances, surface effects reduced
the total $N_r$ signal, which likely resulted from $NH_3$ scavenging to the walls of the transfer liens or quartz tubing. We
conclude that the $N_r$ system with a $CO_2$ detector in parallel can be used as a total carbon measurement system and
would be useful to establish instrument calibrations for carbon-containing aerosol. The system is currently limited to
calibration of compounds in zero air matrices because ambient levels of the common gas-phase carbon compounds
$CO_2$, CO, and $CH_4$ are high.
**3.2 Comparisons with the PILS-ESI/MS**
**3.2.1 $N_r$ system as an aerosol mass measurement method**
Here we demonstrate the capability of the total nitrogen system as an independent calibration method for
aerosol measurement systems. $N_r$ measurements of laboratory generated single-component inorganic and organic
aerosol particles were used to characterize a novel configuration coupling a PILS with electrospray ionization interface
followed by mass spectrometric detection. The strength of using the $N_r$ system to calibrate the PILS-ESI/MS is that it
is a direct method to calibrate the entire coupled on-line system. The current calibration approach involves liquid-
phase standards to calibrate the ESI/MS independently from the PILS.
The inorganic salts selected for this study all contained N atoms, either in the cation, anion, or both. The total
$N_r$ measured as NO (in ppbv) included all the N atoms atomized from the single-component solution. Dividing the
total $N_r$ measurement by the number of N atoms in the parent molecule gives the standard concentration (in ppbv) of
the corresponding anion (e.g. $Cl^-$, $NO_3^-$, $SO_4^{2-}$, $C_2O_4^{2-}$). The mixing ratios (in ppbv) are converted to µg m$^{-3}$ from the





molecular weight of the corresponding anion. We refer to these mass concentrations as "X measured as equivalent $N_r$"
in the remainder of the text, where X is the corresponding anion of the aerosol particle. The anion mass calculated in
this way was only necessary when comparing directly to PILS-ESI/MS measurements of nitrate, sulfate, chloride, and
oxalate.
**3.2.2 $N_r$ and PILS-ESI/MS mass concentration comparisons**
In coupling an aerosol collection technique (PILS) with an electrospray ionization source, water-soluble
aerosol particles are speciated in real-time. To compare to the calibration approaches through liquid phase standards
described in Sect 2.2 for the PILS-ESI/MS we performed particle mass comparisons using these methods with anion-
specific mass concentrations derived from the $N_r$ measurement system. A single-component aerosol was used to
minimize complex matrix effects including ion suppression/enhancement common in ESI.
An example of the $N_r$ system and PILS-ESI/MS co-sampling a laboratory generated polydisperse aerosol
stream is shown in Fig. 8. Here we did not size-select aerosols, but measured all particle sizes below a 2.5 µm cut-off
(URG cyclone, Chapel Hill, NC). There are two reasons for this experimental set-up: (1) Generating a sufficient
aerosol mass concentration to calibrate the PILS-ES/MS was challenging because it requires a minimum flow of 11 L
$min^{-1}$, while the DMA output flow is <1 L $min^{-1}$, therefore the DMA aerosol flow required a large dilution. Because a
greater aerosol particle mass could be realized by directly sampling the polydisperse output of the atomizer, our
analysis focuses on comparisons between $N_r$ and PILS-ESI/MS without using the DMA size-selection. (2)
Conventionally the PILS instrument samples with a cyclone with a 1 or 2.5 µm cutoff, which is similar to other mass
measurement instruments including the aerosol mass spectrometer (AMS) and filter collection.
Figure 8 shows the aerosol nitrate (blue) trace from $NaNO_3$ particles measured by the PILS-ESI/MS shifted
in time to account for the system delay time so that it aligns with the relatively steady concentration periods with the
$N_r$ trace (black). The PILS-ESI/MS had a response time of roughly 4-5 min in its current configuration. Several stages
in the PILS system included mixing volumes (e.g. syringe pumps and mixing vessels) that prevented rapid response
to rapidly changing concentrations and smeared the response. For instrument comparisons 60 s data were averaged
and compared during periods with relatively steady concentrations (generally lasting 5- 10 min). Examples of PILS-
ESI/MS traces aligned such that initial response of both instruments coincide are shown in Fig. S2.
The correlation plot of PILS-ESI/MS to equivalent anion mass measured as $N_r$ for each aerosol-type ($NaNO_3$,
$(NH_4)_2SO_4$, $NH_4Cl$, and $NH_4NO_3$) is shown in Fig. 9(a-d). The concentrations ranged from ~10–120 µg $m^{-3}$ and the
standard linear regression fits for each aerosol type are included in Fig. 9, and were highly correlated with a $R^2 = 0.99$.
For $(NH_4)_2SO_4$, the concentration exceeded the linear dynamic range of the PILS-ESI/MS for sulfate (see Fig. S2(a);
> 130 µg $m^{-3}$) as determined by liquid-standard calibration curves. The linear range of ESI is limited at high
concentrations due to limited surface sites available for ionization (Tang et al., 2004). For this reason values outside
the linear dynamic range of the PILS-ESI/MS ( > 130 µg $m^{-3}$) for sulfate were excluded from the linear regression fit.
$NH_4NO_3$ shows a similar, less pronounced trend, however, it is still included in the regression plot as it was difficult
to isolate whether this was analyte suppression during electrospray ionization or a linear dynamic range issue. Based
on the regression fits in Fig. 9, the difference between the PILS-ESI/MS and $N_r$ system for each inorganic component



is less than 6%. The uncertainty in the ESI signal varies by compound and averaging time, however from the tests
described here the maximum uncertainty is estimated ~15%. Combining this uncertainty with the uncertainty in the
ESI calibrations (maximum ± 10%), the air and liquid flow rate (both ~± 4%), and dilution (~± 5%) in quadrature
gives a total maximum uncertainty associated with mass measurements of ± 20%. So while the slope of the correlations
of the two instruments (based on 60 s averages during periods with constant concentrations) shows a relative difference
of less than ~6%, the uncertainty in the PILS/ESI measurement of single component aerosols is closer to ~ 20% and
could be greater if the transmission and ionization efficiencies of the ESI differ from the efficiencies present during
calibration periods. This uncertainty is greater than the uncertainty (± 10 %) reported for the PILS-IC instrument for
ionic species in Weber et al. (2001) but lower than the AMS uncertainty for nitrate (33 %) and sulfate (35%) estimated
by Bahreini et al. (2009), though the AMS has a much faster time response.

11          Even though greater aerosol particle mass could be produced by directly sampling the polydisperse output of

the atomizer, our analysis also included measurements using the DMA size-selected output. During these tests the
flow was divided between the $N_r$ system, CPC, UHSAS, and PILS-ESI/MS with a large dilution flow that resulted in
turbulent mixing (Re >4000). The CPC and UHSAS particle number concentrations showed improved agreement with
turbulent mixing compared to earlier differences up to 10% at high concentrations discussed in Sect. 2.3 and were
within a few percent of each other. Examples of the real-time temporal profiles for these measurements are shown in
Fig. 10(a-d) with the PILS-ESI/MS time offset by several minutes to account for its delayed response. The calculated
and measured aerosol mass time traces in Fig. 10 show agreement for all measurement techniques tested in this study.
The figures indicate that the PILS-ESI/MS was not given sufficient time to rise to a steady constant concentration for
the first diameter selected. This is confirmed by Fig. 10(b) during which 200 nm particles were size selected twice in
succession with the first selection lasting only ~2 min before flushing with water quickly followed by a longer period
of sampling at the same diameter. The PILS-ESI/MS concentration during this longer sampling period does reach the
expected concentration as indicated by the $N_r$ (black) and CPC (blue) concentrations. The time-series of oxalate in
Fig. 10(d) shows agreement for the equivalent $N_r$ and PILS-ESI/MS measured mass indicating these same calibration
methods are effective for organic compounds, although the UHSAS was not sampling during this experiment. We
conclude that the PILS-ESI/MS quantitatively measures single component inorganic aerosol for a range of sizes,
however, the low particle throughput hindered our ability to evaluate the quantitative abilities of the PILS-ESI/MS
system for particles < 200 nm diameter.

29          These results establish the quantitative abilities of this novel configuration (PILS-ESI/MS) for sampling

simple single-component laboratory generated aerosol. We evaluated this previously uncharacterized mass
measurement technique using both traditional particle number size distribution measuring systems and the total $N_r$
mass measurement system. We show experimentally that the $N_r$ system can be used as a mass calibration method for
pure $N_r$-containing polydisperse aerosol. Calibrating the ESI/MS using direct injection of liquid standards combined
with mass concentrations collected by the PILS is a valid approach for quantifying inorganic components of aerosols,
which likely extends to several organics as demonstrated by oxalate. However, these ESI/MS calibrations are sensitive
to the experimental conditions, which must be precisely maintained during ESI calibrations and throughout the entire
sampling period. Changes in flow rate, interface positioning, or solvent composition have significant impacts on both



the transmission and ionization efficiency ultimately effecting pre-determined ESI calibration factors. Additionally,
PILS characterization has been limited to theoretical predictions or experimental comparisons that involve coupling
the PILS with a mass analyzer (e.g. IC; Orsini et al., 2003; Sorooshian et al., 2006). Here we introduced a new method
for calibrating the entire PILS-ESI/MS coupled system using $N_r$ equivalent mass measurements of $Cl^-$, $NO_3^-$, $SO_4^{2-}$
$,C_2O_4^{2-}$ from $N_r$-containing particles.
**4 Summary and conclusions**
We report the successful application of a total reactive nitrogen ($N_r$) system for conversion of gas-phase and
particle-bound $N_r$-compounds. The $N_r$ system was tested using laboratory-generated monodisperse aerosol from
solutions of $(NH_4)_2SO_4$, $NH_4Cl$, $NaNO_3$, and $(NH_4)_2C_2O_4$. The particle conversion efficiency of each compound was
calculated at each size-selected diameter by the ratio of the concentration measured as $N_r$ to mass concentrations
calculated from number concentration and size distribution measurements using a CPC and UHSAS. Overall, the
particle conversion efficiency for a selection of $N_r$-containing aerosols ranged from 93–101% with an overall estimated
uncertainty of ~10%. The $N_r$- particles tested in these experiments span the range of N oxidation states, and therefore
we are confident these results extend to other $N_r$-containing particles. Most catalyst-based $N_r$ systems measure total
gas-phase $N_r$-only, individual $N_r$-compounds (e.g. $NH_3$), or ignore the contribution of particulate $N_r$ to total signal
completely. However, it is useful to measure the total unspeciated $N_r$ signal, which includes both gases and particles,
to improve our understanding of total N-emissions and their deposition, loss, and availability in ecosystems (e.g.
McCalley and Sparks, 2009). We have presented a rapid, robust measurement technique that quantitatively measures
particle $N_r$ mass that allows for accurately interpreting ambient measurements, and allows improved mass closure of
the N-budget to be constructed for the 2016 Fire Sciences Laboratory measurements of wildfire emissions. Future
applications of this custom system aim to distinguish gas- and particle-phase nitrogen contributions to total measured
$N_r$ signal using upstream filters and denuders.
Additional characterization tests showed the platinum catalyst in the $N_r$ system quantitatively converts both
gaseous- and particulate-organic carbon (OC) to $CO_2$ to within the propagated uncertainties of each detection method
($\pm$ 10% each). The resulting $N_r$ and $C_y$ signals from each detector are in proportion to the number of carbon and
nitrogen atoms in the parent molecule. In order for this to be a reliable total particulate carbon measurement system
under ambient conditions, a highly accurate and precise $CO_2$ measurement system is imperative to measure the signal
above ambient $CO_2$, CO, and $CH_4$ backgrounds. Alternatively, ambient gas-phase constituents could be effectively
eliminated from the sampling matrix. For these reasons, the application of the system is currently limited to calibration
of single-component OC- and/or $N_r$-containing particles.
After establishing efficient conversion of $N_r$-particles, we experimentally demonstrated that this technique
can be used to calibrate aerosol particle mass measurement methods when sampling pure $N_r$-containing polydisperse
aerosol. The $N_r$ equivalent mass measurements of pure atomized polydisperse aerosol showed an agreement of $\pm$ 6%
with the PILS-ESI/MS measurements of the corresponding anion for the salts $(NH_4)_2SO_4$, $NH_4Cl$, $NaNO_3$, and
$NH_4NO_3$. There is a clear advantage to calibrating the entire PILS-ESI/MS system altogether as this avoids
complications arising from calibrating the ESI/MS and PILS independently. We conclude that the $N_r$ system is an





effective measurement technique that can be used to directly calibrate aerosol mass measurement instruments. With
this direct mass calibration method, complications that arise due to optical (e.g. refractive index) and physical
properties (e.g. morphologies) in particle number calibration methods are avoided. Additionally, this method is an on-
line technique that provides a rapid measurement of particle mass unlike off-line mass measurement methods such as
filter analyses. The $N_r$ converter described followed by NO and $CO_2$ detection is a viable new approach for calibrating
aerosol mass instrumentation for both N-containing and organic carbon particles.

**Data availability**

The data from the laboratory tests are available on request. Data from the 2016 Missoula Fire lab are available here:
https://esrl.noaa.gov/csd/groups/csd7/measurements/2016firex/FireLab/DataDownload/index.php?page=/csd/groups
/csd7/measurements/2016firex/FireLab/DataDownload/

**Author contribution**

CES wrote the paper with help from JMR. CES performed the particle calibrations with help from RAW and AK.
JMR and YL built the $N_r$ catalyst and performed the tests to verify gas-phase conversion of $N_r$ species. AM advised
on operation of the PILS. BW, RKT, and CES designed, constructed, and characterized the ESI interface. VS, RJY,
KJZ, CW, and KS made measurements of individual N- species during the FIREX campaign.

**Competing interests**

The authors declare no competing interests or other conflicts of interest.

**Disclaimer**

Mention of commercial products is for identification purposes only and does not imply endorsements.

**Acknowledgements**

This work was supported by NOAA's Climate and Health of the Atmosphere initiatives. C.S. acknowledges support
from the National Research Council Research Associateship Program. B. W. was supported by the Kościuszko
Foundation Program for Advanced Study, Research and/or Teaching in the United States 2014-2015. A. K. is
supported by the Austrian Science Fund FWF's Erwin Schrodinger Fellowship J-3613.The FIREX Fire Lab study
was supported in part by the NOAA Climate Office's Atmospheric Chemistry, Carbon Cycle, and Climate program.
We thank Dr. Brad Hall for the use of his nitrous oxide standard. We thank S. N. Anthony and the Jarrold group for
giving us an ion carpet board for the ESI interface. We thank Matthew Coggon, Abigail Koss, and Joost de Gouw,
for their $H_3O^+$ CIMS data and Steven Brown for his airborne cavity enhanced spectrometer data. We would like to
thank Katherine Manfred, Alessandro Franchin, and Charles Brock for their useful discussions.

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





**Figures**

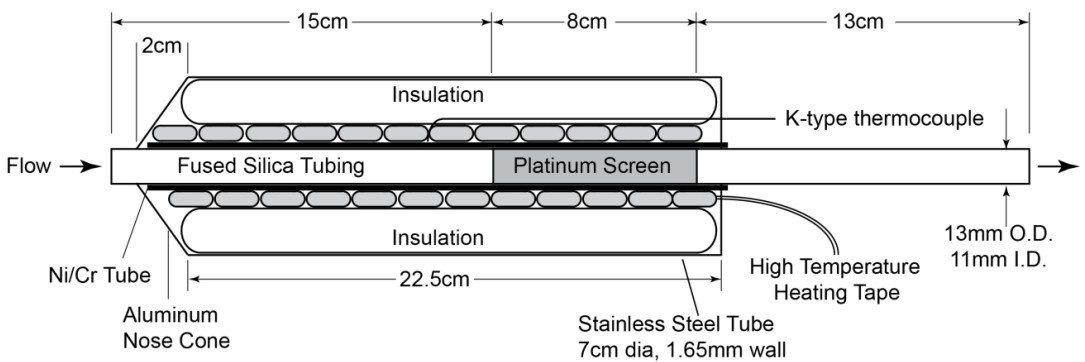

**Figure 1.** Diagram of the custom-built platinum catalyst system for the total reactive nitrogen instrument ($N_r$) operated
at 750°C. The outlet flow is followed by a molybdenum oxide catalyst before the commercial NO-$O_3$
chemiluminescent instrument.



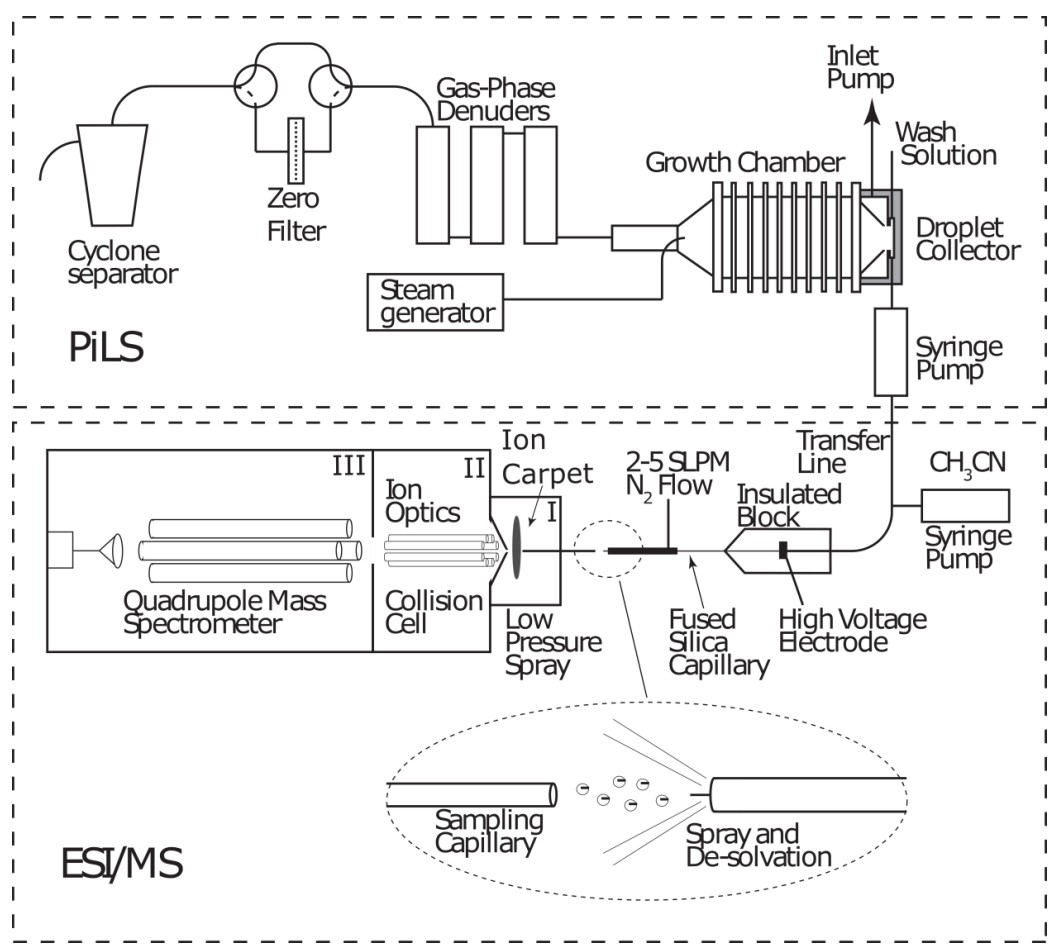

**Figure 2.** Schematic of a particle-into-liquid sampler (PILS; Sorooshian et al., 2006) interfaced to an electrospray ionization (ESI) source of a quadrupole mass spectrometer (MS) for continuous measurement of water soluble components of atmospheric particles.



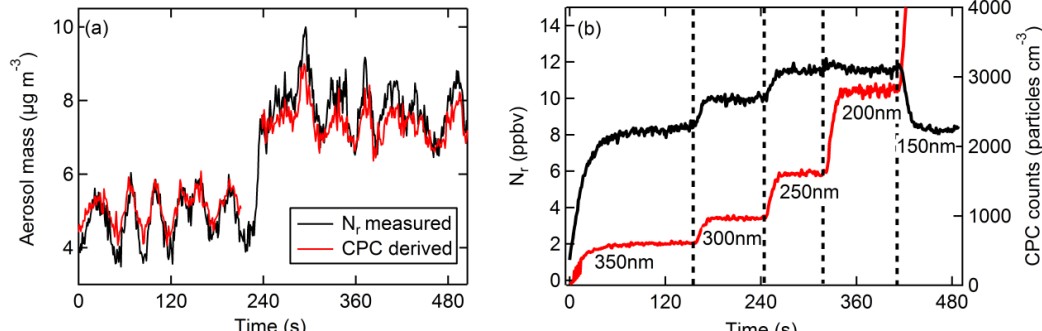

**Figure 3.** The signal resulting from particles only. (a) Real-time $N_r$ (black) measured and CPC (red) derived aerosol mass concentrations ($\mu g\ m^{-3}$) from an atomized solution of $NaNO_3$. (b) Time response of the $N_r$ signal (ppbv) shown in black (left axis), and the CPC signal (particles $cm^{-3}$), shown in red (right axis), as particle sizes of $(NH_4)_2SO_4$ are selectively changed. The dashed vertical lines and labels indicate the singly-charged particle diameter selected with the DMA.



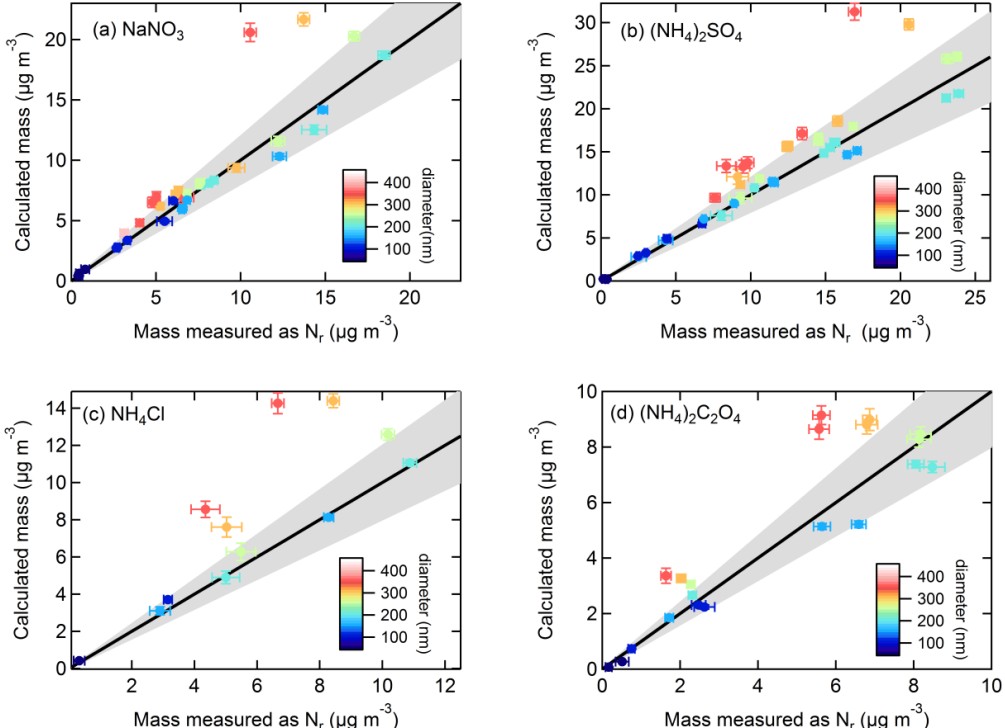

**Figure 4.** Calculated mass from particles size-selected by the DMA and corrected for multiply charged particles using SMPS-derived size distributions compared to aerosol mass concentrations ($\mu$g m$^{-3}$) measured as $N_r$ for (a) NaNO$_3$, (b) (NH$_4$)$_2$SO$_4$, (c) NH$_4$Cl, and (d) (NH$_4$)$_2$C$_2$O$_4$. The particle size is designated by the color plot (error bars indicate $\pm$1 stdev) and the 1:1 line is shown in black with 20% error indicated by the grey shading.





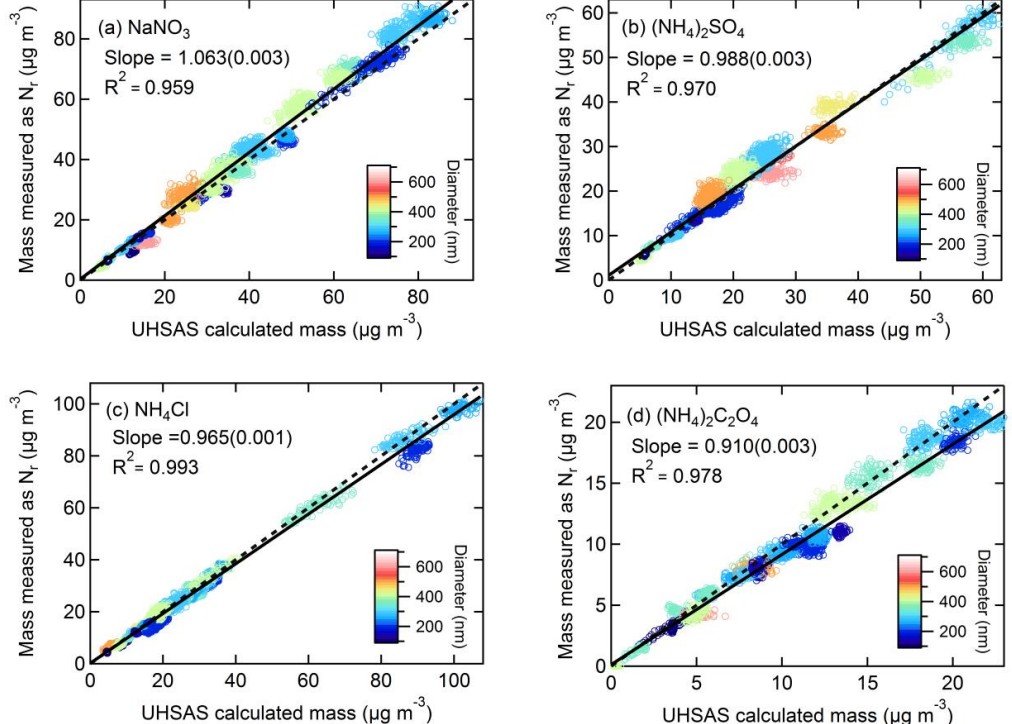

**Figure 5.** Correlation plots of mass concentrations measured as $N_r$ for (a) NaNO$_3$, (b) (NH$_4$)$_2$SO$_4$, (c) NH$_4$Cl, and (d) (NH$_4$)$_2$C$_2$O$_4$) versus mass concentrations calculated using CPC number concentrations with UHSAS size distributions. Particle sizes (nm) are indicated by the color plot and the 1:1 line is shown in dashed black. The solid lines are orthogonal distance regression fits. The slope (uncertainty) and $R^2$ is shown.



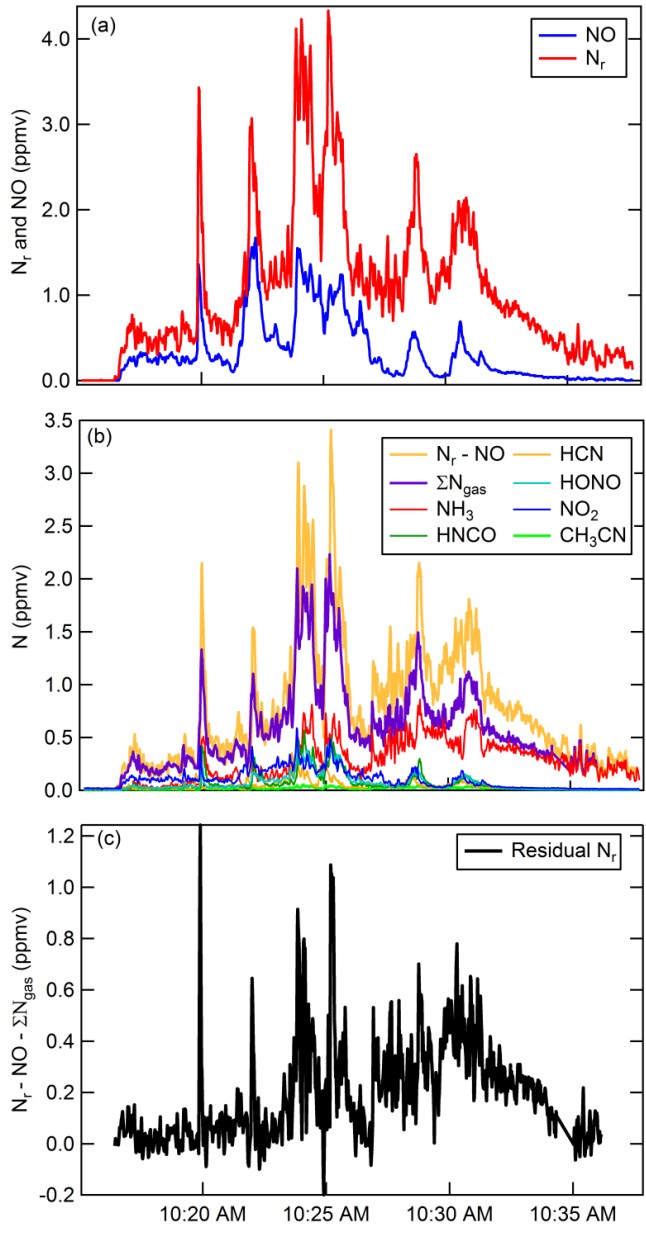

**Figure 6.** Timeseries for Fire Sciences Lab 2016 measurements of emissions from a subalpine fir canopy sample (Fire 047). (a) Total reactive nitrogen ($N_r$, red) and nitric oxide (NO, blue) measurements. (b) Comparison of the difference ($N_r$-NO, gold) with the sum of the measured gas phase $N_r$-species (purple). The sum of individually measured gas-phase species in order of abundance include: $NH_3$, HNCO, HCN, HONO, $NO_2$, $CH_3NO_2$, and 40 minor organic nitrogen species. $NO_2$ and HONO were measured by a broadband cavity enhanced extinction spectrometer, HCN and $NH_3$ were measured by FTIR, and all remaining organic species were measured by $H_3O^+$ CIMS. (c) Residual $N_r$ in ppmv.



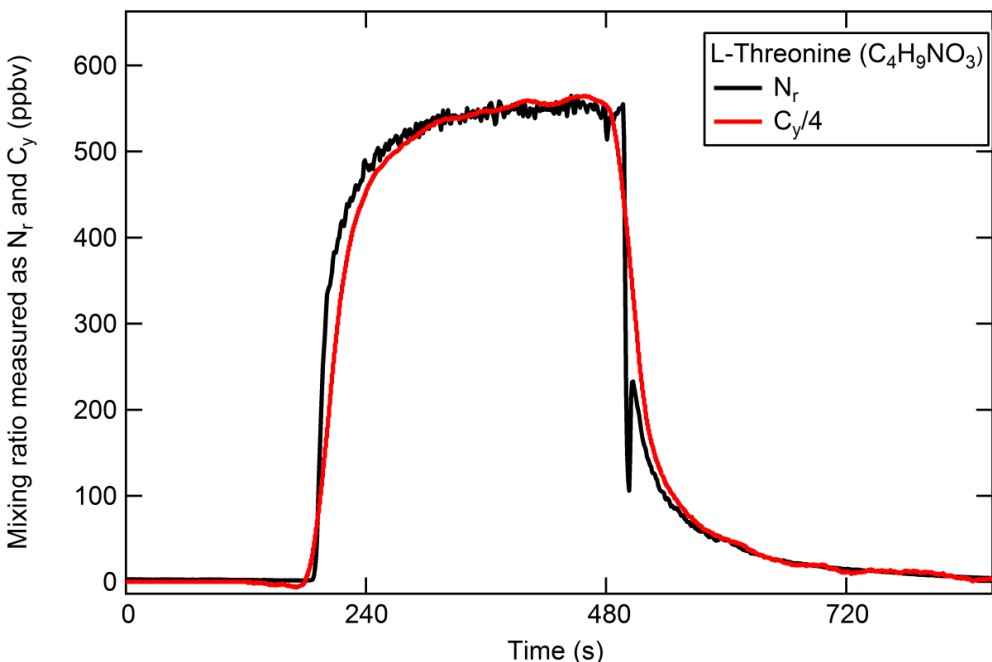

**Figure 7.** An example of the quantitative conversion of atomized polydisperse threonine ($C_4H_9NO_3$) to NO and $CO_2$ measured by NO-$O_3$ chemiluminescence and a LICOR-6251, respectively. The measured total $C_y$ (red) is divided by the number of C atoms in threonine (4).





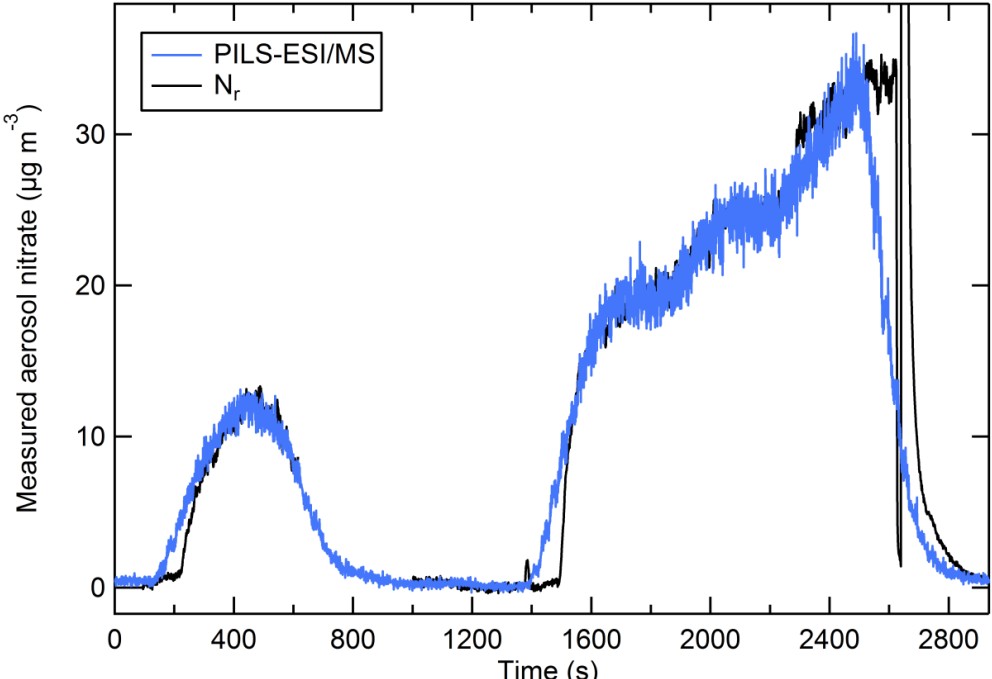

**Figure 8.** The PILS-ESI/MS measured aerosol nitrate mass (blue) and the nitrate measured as $N_r$ (black) ($\mu g\ m^{-3}$) for an atomized solution of $NaNO_3$ (polydisperse). The PILS-ESI/MS trace is shifted to account for the delayed response and the instrument time constant.

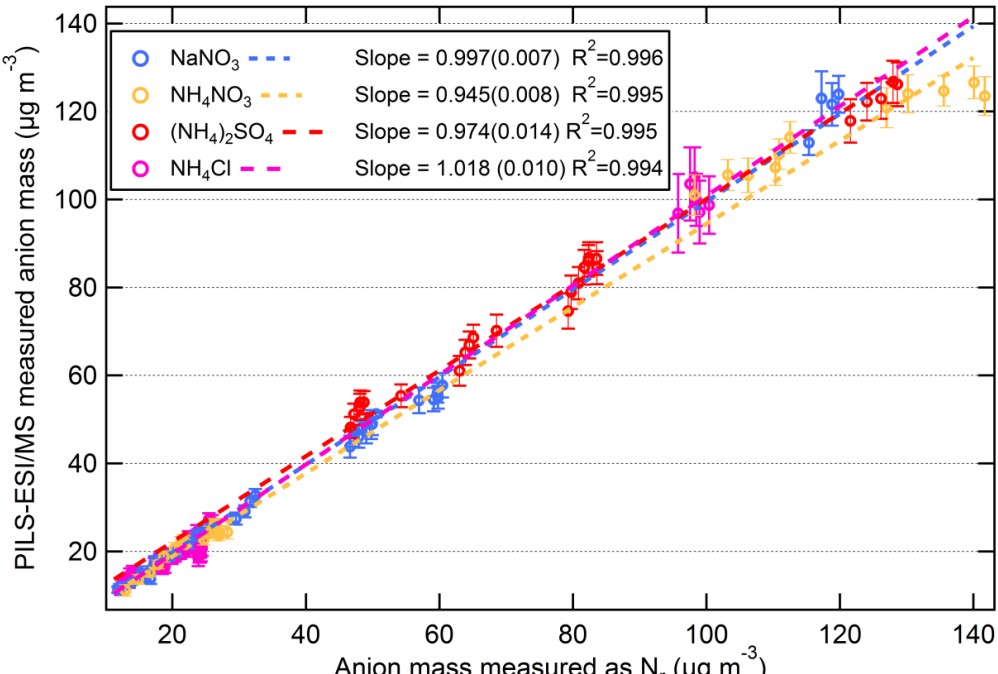

**Figure 9.** Scatter plots of PILS-ESI/MS measured versus equivalent anion mass measured as $N_r$ for salts $NaNO_3$ (blue), $NH_4NO_3$ (gold), $(NH_4)_2SO_4$ (red), and $NH_4Cl$ (magenta). The data are 60 s averages and only include times when the atomized aerosol output was relatively constant (i.e. not when concentrations were rising/falling). The slope (1σ) and $R^2$ is shown.





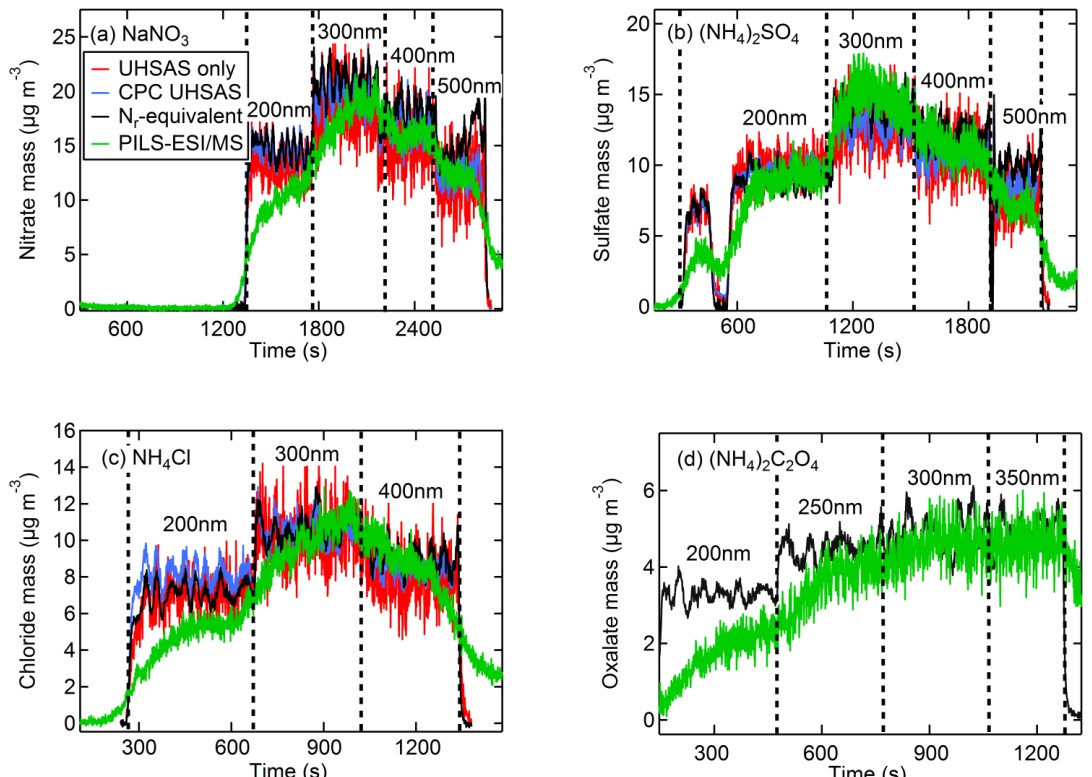

**Figure 10.** The $N_r$ (black) measured, CPC number with UHSAS size (blue) calculated, UHSAS number and size (red) calculated, and PILS-ESI/MS(green) measured aerosol concentrations (µg m⁻³) for anions of DMA size selected aerosol for salts of (a) NaNO₃, (b) (NH₄)₂SO₄, (c) NH₄Cl, and (d) (NH₄)₂C₂O₄. The PILS-ESI/MS traces were shifted in time several minutes early to account for the delayed instrument response time.



**Tables**

**Table 1.** Conversion efficiencies of $N_r$ compounds by the Pt/MoOx catalyst system

| Compound | Conversion efficiency (%) | Calibration method | Reference |
|---|---|---|---|
| Nitrogen Dioxide, $NO_2$ | $99 \pm 2$ | Titration of NO standard by $O_3$ | Williams et al., 1998 |
| Ammonia, $NH_3$ | $105\text{-}110 \pm 15$ | Permeation tube or gas mixture, UV absorbance at 184.9nm | Neuman et al., 2003 |
| Hydrogen cyanide, HCN | $101\text{-}102 \pm 10$ | Gravimetric gas mixture | GASCO, Oldsmar, FL. |
| Cyanogen chloride, ClCN | $98 \pm 10$ | Conversion of HCN standard with Chloramine-T | Valentour et al., 1974 |
| Isocyanic Acid, HNCO | $100 \pm 25$ | Decomposition of the trimer, FTIR | Roberts et al., 2010 |
| Nitrobenzene, $C_6H_5NO_2$ | $95 \pm 15$ | Liquid calibration unit, liquid flow and gravimetric concentration | Ionicon, Innsbruck, Austria |
| Triethyl amine, $(C_2H_5)_3N$ | $95 \pm 15$ | Liquid calibration unit, liquid flow and gravimetric concentration | Ionicon, Innsbruck, Austria |



**Table 2.** Particle conversion efficiencies (%) with uncertainties (one standard deviation) in parentheses. The sizing accuracy is ~± 2.5% using NIST-traceable PSLs for 150 −500 nm spheres as our calibration standard.

| Diameter (nm) | $NaNO_3$ | $(NH_4)_2SO_4$ | $NH_4Cl$ | $(NH_4)_2C_2O_4$ |
|---|---|---|---|---|
| 100 | 88.4(18.3) | 100.6(3.0) | 89.2(5.9) | 91.0(3.5) |
| 150 | 94.0(10.9) | 96.5(2.5) | 93.4(4.7) | 89.0(6.6) |
| 200 | 98.6(4.0) | 98.8(4.8) | 93.6(4.2) | 90.2(5.1) |
| 250 | 101(3) | 100(3) | 98.3(3.7) | 94.7(5.6) |
| 300 | 104(6) | 102(9) | 101(3) | 97.0(6.2) |
| 350 | 102(6) | 101(9) | 98.5(5.2) | 101(13) |
| 400 | 103(8) | 100(8) | 100(6) | 94.7(7.4) |
| 450 | 95.1(4.5) | 110(4) | 103(6) | - |
| 500 | 103(15) | 109(17) | 124(11) | 96.3(7.6) |
| 600 | 83.2(8.7) | 91.9(5.5) | - | 82.5(8.4) |
| Average | 97.3(7.1) | 101(5) | 100(10) | 92.9(5.4) |