# Peer review of "Characterization of a catalyst-based total nitrogen and carbon"

_Atmospheric Measurement Techniques, 2017_

## Referee Comment (RC1) · Anonymous Referee #1 · 2 Feb 2018

In this well written manuscript, the authors present the characterization of a catalyst-based total nitrogen and carbon conversion technique to calibrate particle mass measurement instrumentation, as clearly reflected in the title of the manuscript. Set-up, methodology, and conversion efficiencies for particle-bound nitrogen species are thoroughly discussed. The authors convincingly describe, that the instrument is capable of quantitatively converting a range of particle-bound nitrogen species and provides an online signal of total reactive nitrogen from both gas- and particle-phase, which is very useful for the assessment of nitrogen cycling in the atmosphere. The con-

version of particle-bound carbon via a platinum catalyst is described for a number of organic compounds in laboratory-generated aerosols, while an application to the atmosphere remains challenging due to the small signal compared to background $CO_2$. Nevertheless, a simultaneous detection of total reactive nitrogen and total carbon in one instrumental set-up is a promising perspective. However, the organization of the manuscripts' content could be improved to increase the value of the paper. For example, clearly dividing the subjects instrument characterization (instrument set-up and experiment design, gas-phase $N_r$ conversion efficiency, particle-phase $N_r$ conversion efficiency, particle-phase C conversion efficiency, proof of concept - $N_r$ measurements of biomass burning), and particle mass measurement calibration (laboratory generated aerosols, comparison with PILS-ESI/MS) in sections 2 and 3. A reader could then very quickly see why this new instrument is worth learning about. After addressing content organization and the specific comments listed below, the paper will be very well suited for publication in AMT.

Specific comments:

1. Could you think of a more representative name or acronym for your instrument? The term $N_r$ instrument does not totally reflect the purpose of the instrument in my opinion

2. Please include more recent references on P. 2 L. 6, e.g., Jimenez, et al. 2009, Science; Hallquist et al. 2010, ACP; etc.

3. You sometimes speak of "these experiments" or "these studies" in the manuscript, please consider revising these statements for clarity and readability

4. The purpose of the MoOx catalyst, i.e., reducing $NO_2$ to NO, is not clearly stated in section 2.1

5. Please carefully check through the manuscript again and try to revise extensive and anecdotic paragraphs for conciseness. Exemplarily, please have a look at lines 6 – 29 on page 7 and revise this paragraph.

Exemplary technical comments:

P. 4, L. 3 should read "mass spectrometric detection"

P. 12, L. 20: should read "transfer lines"

P. 13, L. 18: should read "Conventionally,..."

and other small mistakes, which should be considered upon revision of the manuscript

---

## Referee Comment (RC2) · J. Collett (Referee) · 17 Feb 2018

Stockwell et al. report a thorough and satisfying performance evaluation of a catalyst-based approached to measuring particulate reactive N. Although others have explored similar approaches, the work has largely gone unpublished or lacked the thorough evaluation provided by the current authors. There is a compelling need to quantify total reactive N in airborne particles and I commend the authors for their efforts. I also commend them for the thoroughness of their evaluation and the care in which they describe limitations to their approach (e.g., the need to look at particulate OC in a

CO2-free stream, the importance of eliminating PILS-ESI-MS matrix/ion suppression effects by using single component standards, etc....). Their findings will be very useful to the broader atmospheric chemistry community, extending from those interested in source characterization to those interested in deposition and particle effects on human health and radiative scattering. I have a few suggestions for minor changes to improve the manuscript.

1. Title: I found the title confusing and somewhat misleading. The focus is primarily on N and primarily on direct measurement of particulate (or total) reactive N. The title should better reflect that.

2. Abstract: The mention of particulate organic carbon conversion in the abstract is, I suppose, appropriately brief. I do suggest that the authors here refer to "efficient" or "complete" conversion rather than simply conversion. I also suggest they point out here the important challenges of determining particulate OC by this method against a high concentration ambient background, as described in the manuscript itself.

3. Section 3.1.1. The authors refer here to experimental methods not described in the methods section of the manuscript. I suggest an Experimental Details section be added on methods for checking gas-phase conversion efficiency. This would allow the authors to clearly convey information about calibration standards and comparison gas-phase measurement methods. Section 3.1.1, for example, talks about apparent errors in the assumed ammonia absorption cross-section, but this is confusing because the reader has not been told how this is relevant to the gas-phase ammonia measurement method. The latter has not been specified.

4. p. 8, line 29: It seems a bit odd here that the authors refer just to negligible interference from N2O conversion in biomass burning sources. Why only discuss BB and not other (e.g., auto exhaust, ag, etc...) sources. The focus makes a bit more sense given later discussion about the Missoula FIREX experiment, but since this manuscript is really addressing a more broadly applicable approach, it would be helpful to broaden

the N2O interference discussion beyond BB.

5. top of p. 9: It is my sense that it is not so uncommon for NO concentrations to be in the range of 10s of pptv in remote regions. I suggest the authors better justify or moderate their claim that an NO interference of 28 pptv is "clearly a negligible amount in almost any atmospheric context."

6. Section 3.1.4 and Fig. 6. This is an interesting timeline of deriving "excess" reactive N from the new instrument measuring a smoke plume. Do the authors have a measurement of HNO3 in the airstream? I suggest that modified combustion efficiency (MCE) be added as a parameter in Fig. 6, if available, to help make the authors' point re: periods of smoldering vs. flaming combustion.

7. typos:

a. p.6, line 11: change "least-squared" to "least-squares" b. p. 6, line 35: change "promoted" to "promote" p. 12, line 20: change "liens" to "lines"

---

## Author Comment (AC1) · 20 Mar 2018

Author's Comments (AC1) Manuscript: amt-2017-419 Manuscript title: Characterization of a catalyst-based total nitrogen and carbon conversion technique to calibrate particle mass measurement instrumentation

**Response to Reviewers:**

The following discussion includes the reproduced text from the reviewer (bold), along with our detailed responses and the corresponding changes (italics; eliminated text is struck through) made to the revised manuscript. All page and line numbers refer to the original manuscript.

We thank both Referees for their thorough comments and constructive suggestions, which were helpful in improving the manuscript. We have addressed their issues and concerns to the best of our ability.

**Anonymous Referee #1**

In this well written manuscript, the authors present the characterization of a catalyst based total nitrogen and carbon conversion technique to calibrate particle mass measurement instrumentation, as clearly reflected in the title of the manuscript. Set-up, methodology, and conversion efficiencies for particle-bound nitrogen species are thoroughly discussed. The authors convincingly describe, that the instrument is capable of quantitatively converting a range of particle-bound nitrogen species and provides an online signal of total reactive nitrogen from both gas- and particle-phase, which is very useful for the assessment of nitrogen cycling in the atmosphere. The conversion of particle-bound carbon via a platinum catalyst is described for a number of organic compounds in laboratory-generated aerosols, while an application to the atmosphere remains challenging due to the small signal compared to background CO2. Nevertheless, a simultaneous detection of total reactive nitrogen and total carbon in one instrumental set-up is a promising perspective. However, the organization of the manuscripts' content could be improved to increase the value of the paper. For example, clearly dividing the subjects instrument characterization (instrument set-up and experiment design, gas-phase Nr conversion efficiency, particle-phase Nr conversion efficiency, particle-phase C conversion efficiency, proof of concept -Nr measurements of biomass burning), and particle mass measurement calibration (laboratory generated aerosols, comparison with PILS-ESI/MS) in sections 2 and 3. A reader could then very quickly see why this new instrument is worth learning about. After addressing content organization and the specific comments listed below, the paper will be very well suited for publication in AMT.

We appreciate the Reviewer's positive comments, and agree that the manuscript could be reorganized to clarify the experimental approach, motivation, and conclusions. We have divided the manuscript into the following sections: (1) Introduction; (2) Experimental details including (a) instrument descriptions with an added total carbon (Cy) section; (b) experimental design, which includes an added section on methods for determining gas phase conversion efficiency per Reviewer 2's suggestion. The section "particle generation, measurement and characterization" was renamed "methods for determining particle phase conversion efficiency" and a few sentences were added or removed for organizational clarity within this section. (3) Instrument characterization, including both gas and particle conversion efficiency discussions and our "proof of concept" biomass burning emissions measurements; The section discussing the Nr-particle conversion efficiency and (b) Determining Nr-particle conversion efficiency using a DMA and UHSAS. (4) Application to calibrate the PILS-ESI/MS using comparisons with the PiLS-ESI/MS (5) Summary and Conclusions. Sentences throughout the manuscript were occasionally shifted to a new section (see for e.g. the new total carbon (Cy) system section in the experimental details section now incorporates sentences originally included in the results section) or to a more appropriate section to improve the organization as suggested by the Reviewer. While these organizational changes added to the value of this paper, the pages, lines, and a few figure numbers were altered

from the original manuscript. As a result the following changes/discussion following the Reviewer's specific comments will continue to refer to the original manuscript.

The following section was added to the Experimental details:

**"Total carbon (Cy) system**

Measurements of total carbon ( $C_y$ ) were accomplished by catalytic conversion to carbon dioxide ( $CO_2$ ) and detection using a  $CO_2$  analyzer. The high-temperature (750°C), platinum catalyst (Fig. 1) in the  $N_r$  system should quantitatively convert carbon containing species to  $CO_2$  in the presence of air. Gas-phase carbon conversion across similar precious metals has been studied extensively (see for example the Pt catalyst used in Veres et al., 2010). The total flow through the Pt catalyst was set to ~1.5 standard L min-1 and was then split before the MoOx catalyst. In our sampling scheme 0.5 sL min-1 of flow was directed to a LICOR 6251 (LI-6251; Lincoln, NE)  $CO_2$  analyzer, while the remaining flow, 1 sL min-1, was directed through the MoOx catalyst and to the NO-O3 chemiluminescence detector as detailed in Sect. 2.1.1. Run in this manner, the conversion of compounds that contain both N and C atoms can then be measured simultaneously using the NO-O3 chemiluminescence detector and LI-6251 detector in parallel.

The LICOR instrument was internally referenced to scrubbed zero air. At ambient  $CO_2$  levels, it is challenging to retrieve reliable measurements since the signal relative to the background abundance of  $CO_2$  is small. In order to evaluate organic carbon conversion efficiency, our approach relies on using ultra-pure air for aerosol generation and carrier gas flow, therefore ambient CO and  $CO_2$  is eliminated. The LI-6251 was calibrated with sub-5 ppm  $CO_2$  standards (Scott-Marin Inc., Riverside, CA) in ultra-pure air. Due to the low signals levels and the uncertainty of the low concentration  $CO_2$  standards, the overall uncertainty of the  $CO_2$  measurements below 1 ppmv presented in this work is  $\pm 10\%$  for 10 second averages."

**Specific comments:**

**1.** Could you think of a more representative name or acronym for your instrument? The term Nr instrument does not totally reflect the purpose of the instrument in my opinion**

While we appreciate the Reviewer's suggestion to create an alternate acronym this is primarily an instrument for online measurement of gas- and particle-phase total reactive nitrogen. We explicitly state in the introduction that the converter coupled with the NO-O3 chemiluminescence detection is referred to as the "Nr system." For the purposes of organic carbon measurements we direct the sample stream following the heated platinum catalyst to an off-board NDIR CO2 detector and there are additional sampling restrictions since the small signal compared to background CO2 limits ambient sampling. When we discuss organic carbon conversion specifically, we highlight that the method of conversion is across the platinum catalyst only, which is the front-end of our "Nr system." We have also added a subsection to the instrument descriptions section to specifically detail the total carbon measurement approach. Additionally, future experiments will focus on quantifying sulfur conversion followed by SO2 detection and we wish to hold off on naming the complete nitrogen/carbon/sulfur instrument until it is fully characterized.

**2. Please include more recent references on P. 2 L. 6, e.g., Jimenez, et al. 2009, Science; Hallquist et al. 2010, ACP; etc.**

The Reviewer was right to point out that we have not included more recent publications or reviews, thus we have added the two references suggested to the appropriate paragraph.

P2 L7 Added text: "Jimenez et al., 2009; Hallquist et al., 2010"

P19 L10 Added reference: "Hallquist, M., Wenger, J. C., Baltensperger, U., Rudich, Y., Simpson, D., Claeys, M., Dommen, J., Donahue, N. M., George, C., Goldstein, A. H., Hamilton, J. F., Herrmann, H., Hoffmann, T., Iinuma, Y., Jang, M., Jenkin, M. E., Jimenez, J. L., Kiendler-Scharr, A., Maenhaut, W., McFiggans, G., Mentel, Th. F., Monod, A., Prévôt, A. S. H., Seinfeld, J. H., Surratt, J. D., Szmigielski, R., and Wildt, J.: The formation, properties and impact of secondary organic aerosol: current and emerging issues, Atmos. Chem. Phys., 9, 5155-5236, https://doi.org/10.5194/acp-9-5155-2009, 2009."

**3. You sometimes speak of "these experiments" or "these studies" in the manuscript, please consider revising these statements for clarity and readability**

We thank the reviewer for their suggestion and agree that using the phrases "these experiments" and "these studies" are confusing and unnecessary, so we have revised the manuscript in several places by eliminating these phrases as follows:

P4 L11-12 Existing text: "The primary objective of these experiments is to characterize particle conversion" New text: "The primary objectives are to characterize particle conversion"

P4 L34 Existing text: "The operation of this instrument during these experiments often required considerable detuning to keep the instrument count rates below the roll-over point of the photon counting electronics (approximately 5 MHz), thus the detection limit was closer to 0.1 ppbv for these measurements." New text: *"The operation of this instrument often required considerable de-tuning to keep the instrument count rates below the roll-over point of the photon counting electronics (approximately 5 MHz) for the particle concentrations generated, thus the detection limit was closer to 0.1 ppbv (corresponding to 0.3 µg m-3 for aerosol nitrate)."*

P5 L27 Existing text: "A detailed description of the PILS used in these studies can be found" New text: "A detailed description of the PILS can be found"

P12 L33 Existing text: "The inorganic salts selected for this study" New text: "The inorganic salts selected for the comparison between  $N_r$  and the PILS-ESI/MS instruments"

P15 L13 Existing text: "The  $N_r$ - particles tested in these experiments span the" New text: "The  $N_r$ - particles tested span the"

**4. The purpose of the MoOx catalyst, i.e., reducing NO2 to NO, is not clearly stated in section 2.1**

We have added the following sentence:

P4 L31-32 "The heated MoOx catalyst reduces the remaining NO2 to NO."

5. Please carefully check through the manuscript again and try to revise extensive and anecdotic paragraphs for conciseness. Exemplarily, please have a look at lines 6 – 29 on page 7 and revise this paragraph.

While we have edited the suggested paragraph, we feel the information that was not eliminated from the paragraph is important and the organizational changes detailed earlier and completed per Reviewer 1's suggestion more clearly supports their inclusion. The specific revised text is indicated below:

P7 L6-29 Existing text: "In these experiments particle diameters from 100 to 600 nm were selected and the multiplycharged particles in the size distribution were accounted for as described below. For the liquid concentrations and atomizer conditions we used, the DMA output size distribution is a multi-peaked population consisting not only of singly charged particles but also particles with multiple (mostly two or three) charges. The multiply charged particles can contribute significantly to the overall mass and must be considered when calculating particle mass. The distributions of singly, doubly, and triply charged particles can vary depending on the solution concentration. We measured atomized size distributions using the scanning mobility particle sizer (SMPS; Wang and Flagan, 1990) function of the DMA (physical diameter,  $D_p = 1-1000$  nm). The DMA transfer theory (Knutson and Whitby, 1975; Stolzenburg, 1988) with Wiedensohler's (1988) steady-state charge distribution approximation was used to estimate the fraction of multiply charged particles contributing to the CPC number concentration for each diameter setting. There are a number of possible sources of uncertainty using these methods that may include particle losses, DMA transfer function uncertainty, counting uncertainty, and inversion errors. Consequently, the size distribution of particles selected at a particular voltage and flow setting of the DMA was examined using the UHSAS. UHSAS particle sizing is a function of the amount of light scattered onto the photodetectors. The quantity of scattered light, however, depends not only on the particle size, but also on the composition-dependent particle refractive index (Bohren and Huffman, 1983; Liu and Daum, 2000; Hand and Kreidenweis 2002; Rosenberg et al., 2012). The UHSAS manufacturer recommended calibration uses PSL microspheres, which are well characterized and have known refractive index (n = 1.58) and shape. Because the UHSAS sizing is sensitive to particle refractive index, a new sizing calibration curve was produced for each studied particle type (i.e. refractive index) (Kupc et al., 2017). Considering this, we used the DMA, with sizing accuracy  $\sim \pm 2.5\%$  and NIST-traceable PSLs for 150 –500 nm spheres as our calibration standard. The UHSAS sizing was recalibrated by using the DMA to select particles of known size for each of the aerosol types studied. A different UHSAS calibration curve was produced and used for each aerosol type (e.g. Kupc et al., 2017). These calibration curves were used to retrieve accurate particle size distributions so that the multiply charged particles were properly accounted for. "

New text: "For the liquid concentrations, atomizer conditions, and DMA settings used here, the DMA output size distribution was a multi-peaked population consisting not only of singly charged particles but also particles with multiple (mostly two or three) charges that can contribute significantly to the overall particle mass. Hence the particle mass could not be calculated directly from the singly-charged mobility diameter, particle density, and the CPC number concentrations. We generally used two methods to calculate the particle mass concentrations for these experiments. For the first method, the size distributions were measured using the scanning mobility particle sizer (SMPS; Wang and Flagan, 1990) function of the DMA (physical diameter,  $D_p = 1-1000$  nm). We used the DMA transfer theory (Knutson and Whitby, 1975; Stolzenburg, 1988) with Wiedensohler's (1988) steady-state charge distribution approximation to estimate the fraction of multiply charged particles contributing to the CPC number concentration for each diameter setting. There are a number of possible sources of uncertainty using these methods that may include

particle losses, DMA transfer function uncertainty, counting uncertainty, and inversion errors. When comparing mass concentrations from the SMPS with those measured by the  $N_r$  system, issues with the SMPS-derived size distributions became apparent (discussed separately in Section 3.2.2). For the second method of calculating mass concentrations, we directly measured the diluted, DMA output using the UHSAS. UHSAS particle sizing is a function of the amount of light scattered onto the photodetectors, which depends not only on the particle size, but also on the compositiondependent particle refractive index (Bohren and Huffman, 1983; Liu and Daum, 2000; Hand and Kreidenweis 2002; Rosenberg et al., 2012). The UHSAS manufacturer recommended calibration uses PSL microspheres, which are well characterized and have known refractive index (n = 1.58) and shape. Because the UHSAS sizing is sensitive to particle refractive index, a new sizing calibration curve was produced for each studied particle type (i.e. refractive index) using a DMA to select particles for a range of known sizes (Kupc et al., 2018). These calibration curves were used to retrieve accurate particle size distributions that properly accounted for the multiply charged particles. "

The following are additional areas of revised text:

P3 L17 We have eliminated the following text as it is repetitive: "By these methods, Roberts et al. (1998) confirmed efficient conversion of C1-C7 gas-phase compounds across the catalyst."

P3 L30 We have eliminated the following text: "While these instrument calibration techniques are well established for controlled laboratory generated aerosol standards,"

P4 L10-11 Existing text: "a particle-into-liquid sampler coupled directly to an electrospray ionization source and by the Nr instrument." New text: "the PILS-ESI/MS with that measured by the Nr instrument."

P6 L15-16 We eliminated the following text: "For purposes of this comparison, matrix effects were assumed to be negligible for tests sampling single component aerosols."

P7 L35-37 We eliminated the following text: "For these reasons, we used the UHSAS size distributions to estimate the fraction of singly, doubly, and triply charged particles together with the total particle number taken from the CPC measurement to exclusive particle mass from total volume and density." Then we added the following text to the end of the paragraph: "Due to problems with measuring SMPS size distributions and requiring coincidence corrections for the UHSAS number concentrations, we used the UHSAS size distributions with the total particle number taken from the CPC measurement to calculate particle mass from total volume and density."

P10 L10-12 We eliminated the following text: "However, our results demonstrate the added challenges in particle mass determination using estimated size distributions from the SMPS method."

P10 L12 We eliminated the following text: " Other aerosol measurement techniques (e.g. the Particle Time of Flight mode of the Aerosol Mass Spectrometer; DeCarlo et al., 2006) directly measure size distributions or instead measure polydisperse aerosol and the instrument and inversion algorithm corrections required using the SMPS are avoided" And eliminate P18 L8-10: "DeCarlo, P. F., Kimmel, J. R., Trimborn, A., Northway, M. J., Jayne, J. T., Aiken, A. C., Gonin, M., Fuhrer, K., Horvath, T., Docherty, K. S., Worsnop, D. R., and Jimenez, J. L.: Field deployable, high resolution, time of flight aerosol mass spectrometer, Anal. Chem, 78, 8281-8289, doi:10.1021/ac061249n, 2006." P12 L6-7 Existing text: "for the range of oxidation states. We are confident these results extend to other Nr-containing particles, which is supported by the extensive list of Nr gases efficiently converted as shown in Table 1.Therefore, we" New text: "for the range of oxidation states and should extend to other Nr-containing particles, we"

P 12 L18-20 We have eliminated the following text: "*Initial tests with*  $(NH_4)_2C_2O_4$  proved more challenging as the low C number required large polydisperse acrosol loadings (several ppmv) to be measured reliably by the LICOR. During these instances, surface effects reduced the total  $N_r$ -signal, which likely resulted from  $NH_3$ -scavenging to the walls of the transfer lines or quartz tubing." And we combined the remaining sentences following this eliminated text with the above paragraph.

P12 L27-30 Existing text: "Here we demonstrate the capability of the total nitrogen system as an independent calibration method for aerosol measurement systems. Nr measurements of laboratory generated single-component inorganic and organic aerosol particles were used to characterize a novel configuration coupling a PILS with electrospray ionization interface followed by mass spectrometric detection." New text: "*Here we demonstrate the capability of the total nitrogen system as an independent calibration method for other aerosol measurement systems.* Nr measurements of laboratory generated single-component inorganic and organic aerosol particles were used to characterize the PILS-ESI/MS."

P12 L30-32 Existing text: "The strength of using the Nr system to calibrate the PILS-ESI/MS is that it is a direct method to calibrate the entire coupled on-line system. The current calibration approach involves liquid-phase standards to calibrate the ESI/MS independently from the PILS." New text: "*The strength of using the Nr system to calibrate the PILS-ESI/MS and other aerosol mass instruments is that it is a direct method to calibrate the entire coupled on-line system. The current calibration approach for nearly all detectors used with the PILS involves liquid-phase standards to calibrate the detection method independently from the PILS."*

P13 L6 We have eliminated the following text: "In coupling an aerosol collection technique (PILS) with an electrospray ionization source, water soluble aerosol particles are speciated in real time."

P13 L15-17 We have eliminated the following text: "*Because a greater aerosol particle mass could be realized by directly sampling the polydisperse output of the atomizer, our analysis focuses on comparisons between Nr and PILS-ESI/MS without using the DMA size selection.*"

P14 L30-32 We have eliminated the text: "We evaluated this previously uncharacterized mass measurement technique using both traditional particle number size distribution measuring systems and the total Nr mass measurement system." And added text "Here" to begin the following sentence.

P14 L33-35 We have eliminated the following text: "*Calibrating the ESI/MS using direct injection of liquid standards* combined with mass concentrations collected by the PILS is a valid approach for quantifying inorganic components of aerosols, which likely extends to several organics as demonstrated by oxalate."

P15 L2 to P14 L32 We moved the following sentence to an earlier section of the paragraph: "*PILS characterization* has been limited to theoretical predictions or experimental comparisons that involve coupling the PILS with a mass analyzer (e.g. IC; Orsini et al., 2003; Sorooshian et al., 2006)." And we added transition text: "In general,"

P15 L 3-5 We have eliminated the following text: "*Here we introduced a new method for calibrating the entire PILS*-*ESI/MS coupled system using*  $N_r$  equivalent mass measurements of  $Cl^2$ ,  $NO_3^2$ ,  $SO_4^2^2$ ,  $C_2O_4^2^2$  from  $N_r$  containing particles."

P14 L35 We moved the following sentence to an earlier section of the paragraph, existing text: "However, these ESI/MS calibrations are sensitive to the experimental conditions, which must be precisely maintained during ESI calibrations and throughout the entire sampling period. Changes in flow rate, interface positioning, or solvent composition have significant impacts on both the transmission and ionization efficiency ultimately effecting predetermined ESI calibrations, which must be precisely maintained during ESI calibration methods are sensitive to the experimental conditions, which must be precisely maintained during ESI calibrations and throughout the entire sampling period. Changes in flow rate, interface positioning, or solvent composition have significant impacts on both the transmission and uning ESI calibrations and throughout the entire sampling period. Changes in flow rate, interface positioning, or solvent composition have significant impacts on both the transmission and ionization factors."

**Exemplary technical comments:**

**P. 4, L. 3 should read "mass spectrometric detection"**

P4 L3 We added text: "spectrometric" between "mass" and "detection"

**P. 12, L. 20: should read "transfer lines"**

P12 L20 We have changed the existing text: "liens" New text: "lines"

**P. 13, L. 18: should read "Conventionally,: : :"**

P13 L18 We added a comma "," following "Conventionally"

**and other small mistakes, which should be considered upon revision of the manuscript**

There were a few other minor mistakes suggested by Reviewer 2 that we have corrected. Please refer to our response to Reviewer 2 for a few additional corrections. We have included below other small mistakes we have revised in the manuscript.

P2 L27 Existing text: " $O_3$ " New text "ozone ( $O_3$ )"

P3 L22 Existing text: "specifications" New text: "design"

P4 L4 Existing text: "calibrated using" New text: "compared to the calibration obtained with"

P5 L31 Existing text: "The PILS sample flow" New text: "The PILS liquid outlet flow"

P6 L28 Existing text: "atomization of" New text: "atomizing"

P6 L29 Existing text: "in a dry particle-free nitrogen flow" New text: "in a dry particle-free nitrogen or zero air flow"

P6 L29 We added the text: "similar to the one reported by"

P6 L31 We added the text: "similar to one described by"

P6 L33 We eliminated the text: "monodisperse"

P7 L1 We added a comma. Existing text: "output flow following dilution" New text "output flow, following dilution"

P7 L1 Existing text: "We measured the flow " New text: "We measured the CPC flow rate"

P7 L37 Existing text: "exclusive" New text: "calculate"

P9 L15 We inserted additional text: "*on rapid timescales (a few seconds)*" following "However, the total Nr response precisely tracks the CPC signal"

P10 L14 Existing text: "Therefore, we instead" New text: "Here we"

P11 L1 Existing text: "Nr measurements of biomass burning" New text: "Nr measurement of biomass burning emissions"

P11 L33 Existing text: "ambient air is eliminated" New text "ambient CO and CO2 is eliminated"

P12 L21 Existing text: "Nr system" New text: "Nr catalyst"

P13 L3 Existing text: "measurements of nitrate" New text: "measurements of ammonium salts of nitrate"

- P13 L7 Existing text: "through" New text: "using"
- P15 L8 We eliminated the text "monodisperse"

P15 L31 Existing text: "demonstrated that this technique" New text: "demonstrated that the Nr conversion technique"

P15 L33 Existing text: "of" new text: "within"

P16 L28 We eliminated the text "airborne"

- P17 L27 We capitalized the text:"aerodyne"
- P18 L28 Existing text: "P. Natl. Acade." New text: "Proc. Natl. Acad."
- P27 L4 Existing text: "commercial" New text: "custom"

---

## Author Response (AR1)

Author's Comments (AC)
Manuscript: amt-2017-419
Manuscript title: Characterization of a catalyst-based total nitrogen and carbon conversion technique to calibrate particle mass measurement instrumentation

Response to Reviewers:

The following discussion includes the reproduced text from the reviewer (bold), along with our detailed responses and the corresponding changes (italics; eliminated text is struck through) made to the revised manuscript. All page and line numbers refer to the original manuscript.

We thank both Referees for their thorough comments and constructive suggestions, which were helpful in improving the manuscript. We have addressed their issues and concerns to the best of our ability.

Anonymous Referee #1

**In this well written manuscript, the authors present the characterization of a catalyst based total nitrogen and**
**carbon conversion technique to calibrate particle mass measurement instrumentation, as clearly reflected in**
**the title of the manuscript. Set-up, methodology, and conversion efficiencies for particle-bound nitrogen species**
**are thoroughly discussed. The authors convincingly describe, that the instrument is capable of quantitatively**
**converting a range of particle-bound nitrogen species and provides an online signal of total reactive nitrogen**
**from both gas- and particle-phase, which is very useful for the assessment of nitrogen cycling in the atmosphere.**
**The conversion of particle-bound carbon via a platinum catalyst is described for a number of organic**
**compounds in laboratory-generated aerosols, while an application to the atmosphere remains challenging due**
**to the small signal compared to background CO2. Nevertheless, a simultaneous detection of total reactive**
**nitrogen and total carbon in one instrumental set-up is a promising perspective. However, the organization of**
**the manuscripts' content could be improved to increase the value of the paper. For example, clearly dividing**
**the subjects instrument characterization (instrument set-up and experiment design, gas-phase Nr conversion**
**efficiency, particle-phase Nr conversion efficiency, particle-phase C conversion efficiency, proof of concept -**
**Nr measurements of biomass burning), and particle mass measurement calibration (laboratory generated**
**aerosols, comparison with PILS-ESI/MS) in sections 2 and 3. A reader could then very quickly see why this**
**new instrument is worth learning about. After addressing content organization and the specific comments listed**
**below, the paper will be very well suited for publication in AMT.**

We appreciate the Reviewer's positive comments, and agree that the manuscript could be reorganized to clarify the experimental approach, motivation, and conclusions. We have divided the manuscript into the following sections: (1)

Introduction; (2) Experimental details including (a) instrument descriptions with an added total carbon (Cy) section; (b) experimental design, which includes an added section on methods for determining gas phase conversion efficiency per Reviewer 2's suggestion. The section "particle generation, measurement and characterization" was renamed

"methods for determining particle phase conversion efficiency" and a few sentences were added or removed for organizational clarity within this section. (3) Instrument characterization, including both gas and particle conversion efficiency discussions and our "proof of concept" biomass burning emissions measurements; The section discussing the $N_r$-particle conversion efficiency was divided into two subsections (a) Challenges using the DMA/SMPS to determine $N_r$-particle conversion efficiency and (b) Determining $N_r$-particle conversion efficiency using a DMA and

UHSAS. (4) Application to calibrate the PILS-ESI/MS using comparisons with the PiLS-ESI/MS (5) Summary and

Conclusions. Sentences throughout the manuscript were occasionally shifted to a new section (see for e.g. the new total carbon (Cy) system section in the experimental details section now incorporates sentences originally included in the results section) or to a more appropriate section to improve the organization as suggested by the Reviewer. While these organizational changes added to the value of this paper, the pages, lines, and a few figure numbers were altered from the original manuscript. As a result the following changes/discussion following the Reviewer's specific comments will continue to refer to the original manuscript.

The following section was added to the Experimental details:

"*Total carbon (Cy) system*

*Measurements of total carbon ($C_y$) were accomplished by catalytic conversion to carbon dioxide ($CO_2$) and*

*detection using a $CO_2$ analyzer. The high-temperature (750°C), platinum catalyst (Fig. 1) in the $N_r$ system should*

*quantitatively convert carbon containing species to $CO_2$ in the presence of air. Gas-phase carbon conversion across*

*similar precious metals has been studied extensively (see for example the Pt catalyst used in Veres et al., 2010). The*

*total flow through the Pt catalyst was set to ~1.5 standard L min$^{-1}$ and was then split before the MoOx catalyst. In our*

*sampling scheme 0.5 sL min$^{-1}$ of flow was directed to a LICOR 6251 (LI-6251; Lincoln, NE) $CO_2$ analyzer, while the*

*remaining flow, 1 sL min$^{-1}$, was directed through the MoOx catalyst and to the NO-$O_3$ chemiluminescence detector as*

*detailed in Sect. 2.1.1. Run in this manner, the conversion of compounds that contain both N and C atoms can then be*

*measured simultaneously using the NO-$O_3$ chemiluminescence detector and LI-6251 detector in parallel.*

*The LICOR instrument was internally referenced to scrubbed zero air. At ambient $CO_2$ levels, it is*

*challenging to retrieve reliable measurements since the signal relative to the background abundance of $CO_2$ is small.*

*In order to evaluate organic carbon conversion efficiency, our approach relies on using ultra-pure air for aerosol*

*generation and carrier gas flow, therefore ambient CO and $CO_2$ is eliminated. The LI-6251 was calibrated with sub-*

*5 ppm $CO_2$ standards (Scott-Marin Inc., Riverside, CA) in ultra-pure air. Due to the low signals levels and the*

*uncertainty of the low concentration $CO_2$ standards, the overall uncertainty of the $CO_2$ measurements below 1 ppmv*

*presented in this work is ± 10% for 10 second averages.*"

**Specific comments:**

**1. Could you think of a more representative name or acronym for your instrument? The term Nr instrument**
**does not totally reflect the purpose of the instrument in my opinion**

While we appreciate the Reviewer's suggestion to create an alternate acronym this is primarily an instrument for on- line measurement of gas- and particle-phase total reactive nitrogen. We explicitly state in the introduction that the converter coupled with the NO-$O_3$ chemiluminescence detection is referred to as the "$N_r$ system." For the purposes of organic carbon measurements we direct the sample stream following the heated platinum catalyst to an off-board

NDIR $CO_2$ detector and there are additional sampling restrictions since the small signal compared to background $CO_2$

limits ambient sampling. When we discuss organic carbon conversion specifically, we highlight that the method of conversion is across the platinum catalyst only, which is the front-end of our "$N_r$ system." We have also added a subsection to the instrument descriptions section to specifically detail the total carbon measurement approach.

Additionally, future experiments will focus on quantifying sulfur conversion followed by $SO_2$ detection and we wish to hold off on naming the complete nitrogen/carbon/sulfur instrument until it is fully characterized.

**2. Please include more recent references on P. 2 L. 6, e.g., Jimenez, et al. 2009, Science; Hallquist et al. 2010,**
**ACP; etc.**

The Reviewer was right to point out that we have not included more recent publications or reviews, thus we have
added the two references suggested to the appropriate paragraph.

P2 L7 Added text: *"Jimenez et al., 2009; Hallquist et al., 2010"*

P19 L10 Added reference: *"Hallquist, M., Wenger, J. C., Baltensperger, U., Rudich, Y., Simpson, D., Claeys, M.,*
*Dommen, J., Donahue, N. M., George, C., Goldstein, A. H., Hamilton, J. F., Herrmann, H., Hoffmann, T., Iinuma, Y.,*
*Jang, M., Jenkin, M. E., Jimenez, J. L., Kiendler-Scharr, A., Maenhaut, W., McFiggans, G., Mentel, Th. F., Monod,*
*A., Prévôt, A. S. H., Seinfeld, J. H., Surratt, J. D., Szmigielski, R., and Wildt, J.: The formation, properties and impact*
*of secondary organic aerosol: current and emerging issues, Atmos. Chem. Phys., 9, 5155-5236,*
*https://doi.org/10.5194/acp-9-5155-2009, 2009."*

**3. You sometimes speak of "these experiments" or "these studies" in the manuscript, please consider revising**
**these statements for clarity and readability**

We thank the reviewer for their suggestion and agree that using the phrases "these experiments" and "these studies"
are confusing and unnecessary, so we have revised the manuscript in several places by eliminating these phrases as
follows:

P4 L11-12 Existing text: "The primary objective of these experiments is to characterize particle conversion" New text:
*"The primary objectives are to characterize particle conversion"*

P4 L34 Existing text: "The operation of this instrument during these experiments often required considerable de-
tuning to keep the instrument count rates below the roll-over point of the photon counting electronics (approximately
5 MHz), thus the detection limit was closer to 0.1 ppbv for these measurements." New text: *"The operation of this*
*instrument often required considerable de-tuning to keep the instrument count rates below the roll-over point of the*
*photon counting electronics (approximately 5 MHz) for the particle concentrations generated, thus the detection limit*
*was closer to 0.1 ppbv (corresponding to 0.3 µg m$^{-3}$ for aerosol nitrate)."*

P5 L27 Existing text: "A detailed description of the PILS used in these studies can be found" New text: *"A detailed*
*description of the PILS can be found"*

P12 L33 Existing text: "The inorganic salts selected for this study" New text: *"The inorganic salts selected for the*
*comparison between $N_r$ and the PILS-ESI/MS instruments"*

P15 L13 Existing text: "The $N_r$- particles tested in these experiments span the" New text: *"The $N_r$- particles tested*
*span the"*

**4. The purpose of the MoOx catalyst, i.e., reducing NO2 to NO, is not clearly stated in section 2.1**

We have added the following sentence:

P4 L31-32 *"The heated MoOx catalyst reduces the remaining $NO_2$ to NO."*

**5. Please carefully check through the manuscript again and try to revise extensive and anecdotic paragraphs**
**for conciseness. Exemplarily, please have a look at lines 6 – 29 on page 7 and revise this paragraph.**

While we have edited the suggested paragraph, we feel the information that was not eliminated from the paragraph is
important and the organizational changes detailed earlier and completed per Reviewer 1's suggestion more clearly
supports their inclusion. The specific revised text is indicated below:

P7 L6-29 Existing text: **"**In these experiments particle diameters from 100 to 600 nm were selected and the multiply-
charged particles in the size distribution were accounted for as described below. For the liquid concentrations and
atomizer conditions we used, the DMA output size distribution is a multi-peaked population consisting not only of
singly charged particles but also particles with multiple (mostly two or three) charges. The multiply charged particles
can contribute significantly to the overall mass and must be considered when calculating particle mass. The
distributions of singly, doubly, and triply charged particles can vary depending on the solution concentration. We
measured atomized size distributions using the scanning mobility particle sizer (SMPS; Wang and Flagan, 1990)
function of the DMA (physical diameter, $D_p$ = 1-1000 nm). The DMA transfer theory (Knutson and Whitby, 1975;
Stolzenburg, 1988) with Wiedensohler's (1988) steady-state charge distribution approximation was used to estimate
the fraction of multiply charged particles contributing to the CPC number concentration for each diameter setting.
There are a number of possible sources of uncertainty using these methods that may include particle losses, DMA
transfer function uncertainty, counting uncertainty, and inversion errors. Consequently, the size distribution of
particles selected at a particular voltage and flow setting of the DMA was examined using the UHSAS. UHSAS
particle sizing is a function of the amount of light scattered onto the photodetectors. The quantity of scattered light,
however, depends not only on the particle size, but also on the composition-dependent particle refractive index
(Bohren and Huffman, 1983; Liu and Daum, 2000; Hand and Kreidenweis 2002; Rosenberg et al., 2012). The UHSAS
manufacturer recommended calibration uses PSL microspheres, which are well characterized and have known
refractive index (n = 1.58) and shape. Because the UHSAS sizing is sensitive to particle refractive index, a new sizing
calibration curve was produced for each studied particle type (i.e. refractive index) (Kupc et al., 2017). Considering
this, we used the DMA, with sizing accuracy ~ ± 2.5% and NIST-traceable PSLs for 150 −500 nm spheres as our
calibration standard. The UHSAS sizing was recalibrated by using the DMA to select particles of known size for each
of the aerosol types studied. A different UHSAS calibration curve was produced and used for each aerosol type (e.g.
Kupc et al., 2017). These calibration curves were used to retrieve accurate particle size distributions so that the
multiply charged particles were properly accounted for. "

New text: *"For the liquid concentrations, atomizer conditions, and DMA settings used here, the DMA output size*
*distribution was a multi-peaked population consisting not only of singly charged particles but also particles with*
*multiple (mostly two or three) charges that can contribute significantly to the overall particle mass. Hence the particle*
*mass could not be calculated directly from the singly-charged mobility diameter, particle density, and the CPC number*
*concentrations. We generally used two methods to calculate the particle mass concentrations for these experiments.*
*For the first method, the size distributions were measured using the scanning mobility particle sizer (SMPS; Wang*
*and Flagan, 1990) function of the DMA (physical diameter, $D_p$ = 1-1000 nm). We used the DMA transfer theory*
*(Knutson and Whitby, 1975; Stolzenburg, 1988) with Wiedensohler's (1988) steady-state charge distribution*
*approximation to estimate the fraction of multiply charged particles contributing to the CPC number concentration*
*for each diameter setting. There are a number of possible sources of uncertainty using these methods that may include*

*particle losses, DMA transfer function uncertainty, counting uncertainty, and inversion errors. When comparing mass*

*concentrations from the SMPS with those measured by the $N_r$ system, issues with the SMPS-derived size distributions*

*became apparent (discussed separately in Section 3.2.2). For the second method of calculating mass concentrations,*

*we directly measured the diluted, DMA output using the UHSAS. UHSAS particle sizing is a function of the amount*

*of light scattered onto the photodetectors, which depends not only on the particle size, but also on the composition-*

*dependent particle refractive index (Bohren and Huffman, 1983; Liu and Daum, 2000; Hand and Kreidenweis 2002;*

*Rosenberg et al., 2012). The UHSAS manufacturer recommended calibration uses PSL microspheres, which are well*

*characterized and have known refractive index (n = 1.58) and shape. Because the UHSAS sizing is sensitive to particle*

*refractive index, a new sizing calibration curve was produced for each studied particle type (i.e. refractive index)*

*using a DMA to select particles for a range of known sizes (Kupc et al., 2018). These calibration curves were used to*

*retrieve accurate particle size distributions that properly accounted for the multiply charged particles. "*

The following are additional areas of revised text:

P3 L17 We have eliminated the following text as it is repetitive: "

"

P3 L30 We have eliminated the following text: "

"

P4 L10-11 Existing text: "a particle-into-liquid sampler coupled directly to an electrospray ionization source and by the Nr instrument." New text: "*the PILS-ESI/MS with that measured by the $N_r$ instrument.*"

P6 L15-16 We eliminated the following text: "

"

P7 L35-37 We eliminated the following text: "

" Then we added the following text to the end of the paragraph: "*Due to problems with measuring SMPS size distributions and requiring coincidence corrections*

*for the UHSAS number concentrations, we used the UHSAS size distributions with the total particle number taken*

*from the CPC measurement to calculate particle mass from total volume and density.*"

P10 L10-12 We eliminated the following text: "

"

P10 L12 We eliminated the following text: "

"

And eliminate P18 L8-10: "

"

P12 L6-7 Existing text: "for the range of oxidation states. We are confident these results extend to other $N_r$-containing particles, which is supported by the extensive list of $N_r$ gases efficiently converted as shown in Table 1.Therefore, we" New text: "*for the range of oxidation states and should extend to other $N_r$-containing particles, we*"

P 12 L18-20 We have eliminated the following text: "*Initial tests with $(NH_4)_2C_2O_4$ proved more challenging as the*

*low C number required large polydisperse aerosol loadings (several ppmv) to be measured reliably by the LICOR.*

*During these instances, surface effects reduced the total $N_r$ signal, which likely resulted from $NH_3$ scavenging to the*

*walls of the transfer lines or quartz tubing.*" And we combined the remaining sentences following this eliminated text with the above paragraph.

P12 L27-30 Existing text: "Here we demonstrate the capability of the total nitrogen system as an independent calibration method for aerosol measurement systems. $N_r$ measurements of laboratory generated single-component inorganic and organic aerosol particles were used to characterize a novel configuration coupling a PILS with electrospray ionization interface followed by mass spectrometric detection." New text: "*Here we demonstrate the*

*capability of the total nitrogen system as an independent calibration method for other aerosol measurement systems.*

*$N_r$ measurements of laboratory generated single-component inorganic and organic aerosol particles were used to*

*characterize the PILS-ESI/MS.*"

P12 L30-32 Existing text: "The strength of using the $N_r$ system to calibrate the PILS-ESI/MS is that it is a direct method to calibrate the entire coupled on-line system. The current calibration approach involves liquid-phase standards to calibrate the ESI/MS independently from the PILS." New text: "*The strength of using the $N_r$ system to*

*calibrate the PILS-ESI/MS and other aerosol mass instruments is that it is a direct method to calibrate the entire*

*coupled on-line system. The current calibration approach for nearly all detectors used with the PILS involves liquid-*

*phase standards to calibrate the detection method independently from the PILS.*"

P13 L6 We have eliminated the following text: "*In coupling an aerosol collection technique (PILS) with an*

*electrospray ionization source, water-soluble aerosol particles are speciated in real-time.*"

P13 L15-17 We have eliminated the following text: "*Because a greater aerosol particle mass could be realized by*

*directly sampling the polydisperse output of the atomizer, our analysis focuses on comparisons between $N_r$ and PILS-*

*ESI/MS without using the DMA size-selection.*"

P14 L30-32 We have eliminated the text: "*We evaluated this previously uncharacterized mass measurement technique*

*using both traditional particle number size distribution measuring systems and the total $N_r$ mass measurement*

*system.*" And added text "*Here*" to begin the following sentence.

P14 L33-35 We have eliminated the following text: "*Calibrating the ESI/MS using direct injection of liquid standards*

*combined with mass concentrations collected by the PILS is a valid approach for quantifying inorganic components*

*of aerosols, which likely extends to several organics as demonstrated by oxalate.*"

P15 L2 to P14 L32 We moved the following sentence to an earlier section of the paragraph: "*PILS characterization*

*has been limited to theoretical predictions or experimental comparisons that involve coupling the PILS with a mass*

*analyzer (e.g. IC; Orsini et al., 2003; Sorooshian et al., 2006).*" And we added transition text: "*In general,*"

P15 L 3-5 We have eliminated the following text: ""

P14 L35 We moved the following sentence to an earlier section of the paragraph, existing text: "However, these ESI/MS calibrations are sensitive to the experimental conditions, which must be precisely maintained during ESI calibrations and throughout the entire sampling period. Changes in flow rate, interface positioning, or solvent composition have significant impacts on both the transmission and ionization efficiency ultimately effecting pre-determined ESI calibration factors." New text: "*However, our current ESI/MS calibration methods are sensitive to the experimental conditions, which must be precisely maintained during ESI calibrations and throughout the entire sampling period. Changes in flow rate, interface positioning, or solvent composition have significant impacts on both the transmission and ionization efficiency ultimately effecting pre-determined ESI calibration factors.*"

**Exemplary technical comments:**

**P. 4, L. 3 should read "mass spectrometric detection"**

P4 L3 We added text: *"spectrometric"* between "mass" and "detection"

**P. 12, L. 20: should read "transfer lines"**

P12 L20 We have changed the existing text: "liens" New text: *"lines"*

**P. 13, L. 18: should read "Conventionally,: : :"**

P13 L18 We added a comma "," following "Conventionally"

**and other small mistakes, which should be considered upon revision of the manuscript**

There were a few other minor mistakes suggested by Reviewer 2 that we have corrected. Please refer to our response to Reviewer 2 for a few additional corrections. We have included below other small mistakes we have revised in the manuscript.

P2 L27 Existing text: "$O_3$" New text "*ozone ($O_3$)*"

P3 L22 Existing text: "specifications" New text: "design"

P4 L4 Existing text: "calibrated using" New text: "*compared to the calibration obtained with*"

P5 L31 Existing text: "The PILS sample flow" New text: "*The PILS liquid outlet flow*"

P6 L28 Existing text: "atomization of" New text: "atomizing"

P6 L29 Existing text: "in a dry particle-free nitrogen flow" New text:"*in a dry particle-free nitrogen or zero air flow*"

P6 L29 We added the text: "*similar to the one reported by*"

P6 L31 We added the text: "*similar to one described by*"

P6 L33 We eliminated the text: ""

P7 L1 We added a comma. Existing text: "output flow following dilution" New text "*output flow, following dilution*"

P7 L1 Existing text: "We measured the flow " New text: "*We measured the CPC flow rate*"

P7 L37 Existing text: "exclusive" New text: "*calculate*"

P9 L15 We inserted additional text: "*on rapid timescales (a few seconds)"* following "However, the total Nr response precisely tracks the CPC signal"

P10 L14 Existing text: "Therefore, we instead" New text: "*Here we*"

P11 L1 Existing text: "$N_r$ measurements of biomass burning" New text: "$N_r$ measurement of biomass burning emissions"

P11 L33 Existing text: "ambient air is eliminated" New text "*ambient CO and $CO_2$ is eliminated*"

P12 L21 Existing text: "$N_r$ system" New text: "*$N_r$ catalyst*"

P13 L3 Existing text: "measurements of nitrate" New text: "*measurements of ammonium salts of nitrate*"

P13 L7 Existing text: "through" New text: "*using*"

P15 L8 We eliminated the text ""

P15 L31 Existing text: "demonstrated that this technique" New text: "*demonstrated that the $N_r$ conversion technique*"

P15 L33 Existing text: "of" new text: "*within*"

P16 L28 We eliminated the text ""

P17 L27 We capitalized the text:"aerodyne"

P18 L28 Existing text: "P. Natl. Acade." New text: "*Proc. Natl. Acad.*"

P27 L4 Existing text: "commercial" New text: "*custom*"

**J.Collett (Referee 2)**

**Stockwell et al. report a thorough and satisfying performance evaluation of a catalyst based approached to**
**measuring particulate reactive N. Although others have explored similar approaches, the work has largely gone**
**unpublished or lacked the thorough evaluation provided by the current authors. There is a compelling need to**
**quantify total reactive N in airborne particles and I commend the authors for their efforts. I also commend**
**them for the thoroughness of their evaluation and the care in which they describe limitations to their approach**
**(e.g., the need to look at particulate OC in a CO2-free stream, the importance of eliminating PILS-ESI-MS**
**matrix/ion suppression effects by using single component standards, etc....). Their findings will be very useful**
**to the broader atmospheric chemistry community, extending from those interested in source characterization**
**to those interested in deposition and particle effects on human health and radiative scattering. I have a few**
**suggestions for minor changes to improve the manuscript.**

We thank Dr. Jeffrey Collett for his positive review and useful comments, which have added value to this paper.

Additional specific responses are included below.

**1. Title: I found the title confusing and somewhat misleading. The focus is primarily on N and primarily on**
**direct measurement of particulate (or total) reactive N. The title should better reflect that.**

We agree that it is important to straightforwardly describe the focus of this paper in the title. While we are characterizing a catalyst approach to quantitatively convert particulate nitrogen and organic carbon, we also believe it is important to present comparisons to other mass measurement systems as this may be of interest to scientists interested in alternate calibration approaches for their particle mass measurement systems. Thus, we have changed the title to better describe the manuscript

Existing title: "Characterization of a catalyst-based total nitrogen and carbon conversion technique to calibrate particle mass measurement instrumentation"

New title: *"Characterization of a catalyst-based conversion technique to measure total particulate nitrogen and*

*organic carbon and comparison to a particle mass measurement instrument "*

**2. Abstract: The mention of particulate organic carbon conversion in the abstract is, I suppose, appropriately**
**brief. I do suggest that the authors here refer to "efficient" or "complete" conversion rather than simply**
**conversion. I also suggest they point out here the important challenges of determining particulate OC by this**
**method against a high concentration ambient background, as described in the manuscript itself.**

We have added "efficient" before "conversion" and briefly describe this method's shortcomings for application to ambient sampling of particulate carbon.

P1 L24 We added text: "*efficient*" before "conversion"

P1 L25 We added new text to the abstract: "*However, the application of this method to the atmosphere presents a*

*challenge due to the small signal above background at high ambient levels of common gas-phase carbon compounds*

*(e.g. $CO_2$)."*

**3. Section 3.1.1. The authors refer here to experimental methods not described in the methods section of the**
**manuscript. I suggest an Experimental Details section be added on methods for checking gas-phase conversion**
**efficiency. This would allow the authors to clearly convey information about calibration standards and**
**comparison gas phase measurement methods. Section 3.1.1, for example, talks about apparent errors in the**
**assumed ammonia absorption cross-section, but this is confusing because the reader has not been told how this**
**is relevant to the gas-phase ammonia measurement method. The latter has not been specified.**

We have added a section to more clearly describe the gas-phase conversion experiments. Per Reviewer 1's suggestions we have reorganized the manuscript, thus this new section was added to the "Experimental design" section prior to our discussion focusing on methods for determining particle phase conversion efficiency.

New added text/section:

"*2.2.1 Methods for determining gas-phase conversion efficiency*

*"The efficiency of conversion of several N-containing gases by the $N_r$ catalyst was determined through*

*addition of a number of representative compounds that were calibrated independently. The NO signal from the*

*converted species was then compared to the signal from an NO in $N_2$ standard (5.38 ppmv, Scott-Marrin Inc.,*

*Riverside, CA) that was used as the working standard for this project. Typical calibration levels were in the range of*

*50 to 100 ppbv as determined by the mass flow controllers used to mix the standard into the measurement stream. The*

*standards used for each compound and their associated calibrations were as follows.*

*The $NO_2$ standard stream was produced from the NO working standard through gas-phase titration with a*

*small stream of $O_2$ in which $O_3$ had been produced by photolysis at 184.9 nm using a mercury discharge lamp. This*

*technique is used routinely for $NO_x$ and $NO_y$ measurement systems (Williams et al., 1998) and allows straightforward*

*determination of $NO_2$ conversion provided care is taken not to over-titrate the NO stream to produce $NO_3$ and*

*therefore $N_2O_5$. The uncertainty in the $NO_2$ conversion determination is simply the propagated errors in the*

*subtraction of the signals before and after titration.*

*The ammonia ($NH_3$) conversion was examined using two different $NH_3$ sources, a gas mixture (3.1 ppmv in*

*$N_2$, Scott-Marrin) and a permeation device (Kin-Tek, LaMarque, Texas). Care was taken with these standards to keep*

*them under flow for periods of several days in order to insure any system surfaces were equilibrated. The calibration*

*of these standards was accomplished by ultraviolet (UV) absorption spectroscopy at 184.9 nm wavelength using an*

*instrument described by Neuman et al, (2003), and based on absorption cross-sections reported in the literature*

*(Tannenbaum et al., 1953; Lovejoy, 1999; Froyd, 2002). The uncertainty of $NH_3$ conversion was propagated based*

*on the uncertainties in flow rate and UV absorption determinations.*

*The hydrogen cyanide (HCN) standard consisted of a commercial gravimetric mixture of HCN in $N_2$ (10*

*ppmv, GASCO Oldsmar, FL), which was mixed into the system using a mass flow controller. The specified uncertainty*

*of this mixture was ±10%, and the standard concentration was verified using long-path Fourier-transform infrared*

*(FTIR) spectroscopy to within the stated uncertainty. The HCN standard was used to produce a gas phase stream of*

*cyanogen chlorine (ClCN) by reaction with chloramine-T, a non-volatile chlorinating agent, which has been described*

*previously (Valentour et al., 1974). To do this, a small stream (5 standard $cm^3 min^{-1}$) of the HCN standard was*

*combined with humidified Zero Air (ZA, 60% RH, 30 standard $cm^3 min^{-1}$) over a bed packed with glass beads coated*

*with a solution of chloramine-T. The glass beads were prepared by coating glass 3 mm outer diameter (OD) beads*

*with a 2 g 100 $mL^{-1}$ solution and packing ~20 $cm^3$ of them in a 12.7 mm OD PFA tube and flowing ZA over them until*

*dry. The reaction was shown to be essentially 100% (±10%) by proton-transfer reaction mass spectrometry (PTR-*

*MS) when conducted in a humidified atmosphere (RH ≥60%), by FTIR analysis of the HCN and ClCN in the gas*

*stream before and after chlorination.*

*The isocyanic acid (HNCO) standard was prepared according to the methods described by Roberts et al.*

*(2010), in which the trimer, cyanuric acid, was thermally decomposed at 250°C in a diffusion cell to produce a steady*

*stream of HNCO, which was then calibrated by long-path FTIR spectroscopy. Initially, this source has the potential*

*to produce $NH_3$ as an impurity, most likely because of the presence of trace amounts of water. Keeping the source*

*under flow and above 120°C at all times when not in use was found to reduce the $NH_3$ impurity to negligible levels*

*(<5%), as measured by PTR-MS. The uncertainties in the HNCO standard were propagated from the uncertainties in*

*the HNCO cross section (Northwest-Infrared, PNNL), the $NH_3$ subtraction, and flow rates. Standard streams of both*

*nitrobenzene and trimethylamine were produced using gravimetrically prepared solutions and a commercial liquid*

*calibration device (Ionicon, Innsbruck, Austria). The uncertainties in these liquid calibration standards were*

*estimated from the propagated uncertainties in the solution concentrations and the liquid and gas flow rates*

*The conversion of nitrous oxide ($N_2O$) is a potential interference in the $N_r$ method as $N_2O$ is not typically*

*considered a reactive nitrogen compound in the troposphere. Several experiments were conducted to determine the*

*extent of this potential interference using a 10.1 ppmv $N_2O$ standard. The resulting conversion efficiency ranged from*

*0.03% to 0.05% in dry and humidified air respectively. These can be considered upper limits for this interference as*

*we cannot be completely sure that there were no $N_r$ contaminants (e.g. $NO_2$) in the $N_2O$ standard."*

We added the following references to accompany the above text:

P19 L3 Added text: "*Froyd, K. D.: Ion induced nucleation in the atmosphere: Studies of ammonia, sulfuric acid, and*

*water cluster ions, Ph.D., Department of Chemistry, University of Colorado, Boulder, Colorado, 282 pp., 2002.*"

P21 L29 Added text: "*Lovejoy, E. R.: Ion trap studies of H+(H2SO4)m(H2O)n reactions with water, ammonia, and*

*a variety of organic compounds, Int. J. Mass Spectrom., 190/191, 231-241, 1999.*"

P25 L1 Added text: "*Tannenbaum, E., Coffin, E. M., and Harrison, A. J.: The far ultraviolet absorption spectra of*

*simple alkyl amines, J. Chem. Phys., 21, 311, doi: https://doi.org/10.1063/1.1698878, 1953.*"

P8 L6-7 Existing text: "We verified the efficiency of conversion of a range of gas phase $N_r$ compounds in this catalyst system using calibrated gas mixtures or standard streams and auxiliary analysis methods." New text: "*We verified the*

*efficiency of conversion of a range of gas phase $N_r$ compounds in this catalyst system using calibrated gas mixtures*

*or standard streams and auxiliary analysis methods as described in Sect. 2.2.1.*"

P8 L7-9 We eliminated thr text: "

"

P8 L11-12 Existing text: "The uncertainties in the measured conversion efficiencies encompass the propagated errors in each calibration method." New text: "*The uncertainties in the measured conversion efficiencies are the propagated*

*errors in each calibration method, and in all cases the range encompasses 100 % conversion*"

**4. p. 8, line 29: It seems a bit odd here that the authors refer just to negligible interference from N2O conversion**

**in biomass burning sources. Why only discuss BB and not other (e.g., auto exhaust, ag, etc...) sources. The focus**

**makes a bit more sense given later discussion about the Missoula FIREX experiment, but since this manuscript**

**is really addressing a more broadly applicable approach, it would be helpful to broaden the N2O interference discussion beyond BB**

We have added the following text to extend the $N_2O$ discussion to other sources so that biomass burning is not implied as the only important source mentioned in the manuscript:

P8 L30 New text added: "*$N_2O$ emissions from other sources (e.g. natural and anthropogenic agricultural sources,*

*fossil fuel combustion, or animal waste) can be significant, therefore the interference from $N_2O$ conversion must be*

*considered*"

**5. top of p. 9: It is my sense that it is not so uncommon for NO concentrations to be in the range of 10s of pptv in remote regions. I suggest the authors better justify or moderate their claim that an NO interference of 28 pptv is "clearly a negligible amount in almost any atmospheric context."**

The Reviewer raises a valid point here, which we address below.

P9 L1-2 Existing text: "an upper limit that is clearly a negligible amount in almost any atmospheric context" New text: "*an upper limit that is generally a negligible amount in almost any atmospheric context except in more remote*

*regions.*"

**6. Section 3.1.4 and Fig. 6. This is an interesting timeline of deriving "excess" reactive N from the new instrument measuring a smoke plume. Do the authors have a measurement of HNO3 in the airstream? I suggest that modified combustion efficiency (MCE) be added as a parameter in Fig. 6, if available, to help make the authors' point re: periods of smoldering vs. flaming combustion.**

$HNO_3$ was not measured during the experiment though we'd expect very low concentrations from biomass burning as much of the $HNO_3$ formed likely reacts quickly with $NH_3$ to form particle nitrate (Yokelson et al., 2009; see reference below). The Reviewer brings up a useful suggestion to add MCE to the figure to better support the differences in emissions between smoldering and. flaming, therefore we have added MCE to panel (c) of Fig. 6 and the following additions to the text:

See: Yokelson, R. J., Crounse, J. D., DeCarlo, P. F., Karl, T., Urbanski, S., Atlas, E., Campos, T., Shinozuka, Y.,

Kapustin, V., Clarke, A. D., Weinheimer, A., Knapp, D. J., Montzka, D. D., Holloway, J., Weibring, P., Flocke, F.,

Zheng, W., Toohey, D., Wennberg, P. O., Wiedinmyer, C., Mauldin, L., Fried, A., Richter, D., Walega, J., Jimenez,

J. L., Adachi, K., Buseck, P. R., Hall, S. R., and Shetter, R.: Emissions from biomass burning in the Yucatan, Atmos.

Chem. Phys., 9, 5785-5812, https://doi.org/10.5194/acp-9-5785-2009, 2009.

P11 L15 Added text: "*The modified combustion efficiency (MCE) is a measure to estimate the relative contribution*

*of flaming and smoldering combustion that occurred over the course of a fire, where the MCE is defined as the ratio*

*of $\Delta CO_2 / (\Delta CO_2 + \Delta CO)$ (Yokelson et al., 1996). A higher MCE value (approaching 0.99) designates relatively pure*

*flaming combustion (more complete oxidation) and a lower MCE ($\sim$ 0.75–0.84) designates more smoldering*

*combustion.*"

P11 L16-17 Existing text: "it is likely that particulate ammonium contributes to the excess $N_r$ signal measured during periods dominated by smoldering combustion" New text: "*it is possible that the residual signals are due to particulate*

*N-containing compounds. Particulate ammonium may contribute to the excess $N_r$ signal measured during periods*

*dominated by smoldering combustion (MCE < 0.90).*"

P11 L17-18 Existing text: ", while particulate nitrate likely accounts for some $N_r$ signal during the flaming dominated
stages as shown in Fig. 6." New text: "*The oxidized N-containing gas phase species are relatively more abundant*
*during the initial part of the fire, so particulate nitrate could account for some $N_r$ signal during the flaming dominated*
*stages as shown in Fig. 6.*"

P32 Figure 6 caption; Existing text: "(c) Residual Nr in ppmv" New text: "*(c) Residual $N_r$ (black) in ppmv with*
*modified combustion efficiency overlaid (MCE, red).*"

P25 L3 Add reference "*Yokelson, R. J., Griffith, D. W. T., and Ward, D. E.: Open path Fourier transform infrared*
*studies of large-scale laboratory biomass fires, J. Geophys. Res., 101, 21067–21080, doi:10.1029/96jd01800, 1996.*"

Please see updated Figure below:

[Figure]

**7. typos: a. p.6, line 11: change "least-squared" to "least-squares" b. p. 6, line 35: change "promoted" to "promote" p. 12, line 20: change "liens" to "lines"**

Thank you for bringing these minor mistakes to our attention. We have corrected the mistakes as follows:

P6 L11 Existing text: "least-squared" New text: "*least-squares*"

P6 L35 Existing text: "promoted" New text: "*promote*"

P12 L20 Existing text: "liens" New text: "*lines*"

The following includes voluntary changes to references, which includes several updates

[revised manuscript text omitted]

**2.1.2 Total carbon ($C_y$) system**

Measurements of total carbon (Cy) were accomplished by catalytic conversion to carbon dioxide ($CO_2$) and detection using a $CO_2$ analyzer. The high-temperature (750°C), platinum catalyst (Fig. 1) in the $N_y$ system should quantitatively convert carbon containing species to $CO_2$ in the presence of air. Gas-phase carbon conversion across similar precious metals has been studied extensively (see for example the Pt catalyst used in Veres et al., 2010). The total flow through the Pt catalyst was set to ~1.5 standard L min$^{-1}$ and was then split before the MoOx catalyst. In our sampling scheme 0.5 sL min$^{-1}$ of flow was directed to a LICOR 6251 (LI-6251; Lincoln, NE) $CO_2$ analyzer, while the remaining flow, 1 sL min$^{-1}$, was directed through the MoOx catalyst and to the NO-$O_3$ chemiluminescence detector as detailed in Sect. 2.1.1. Run in this manner, the conversion of compounds that contain both N and C atoms can then be measured simultaneously using the NO-$O_3$ chemiluminescence detector and LI-6251 detector in parallel.

The LICOR instrument was internally referenced to scrubbed zero air. At ambient $CO_2$ levels, it is challenging to retrieve reliable measurements since the signal relative to the background abundance of $CO_2$ is small. In order to evaluate organic carbon conversion efficiency, our approach relies on using ultra-pure air for aerosol generation and carrier gas flow, therefore ambient CO and $CO_2$ is eliminated. The LI-6251 was calibrated with sub-5 ppm $CO_2$ standards (Scott-Marin Inc., Riverside, CA) in ultra-pure air. Due to the low signals levels and the uncertainty of the low concentration $CO_2$ standards, the overall uncertainty of the $CO_2$ measurements below 1 ppmv presented in this work is ± 10% for 10 second averages.

**2.1.3 PILS-ESI/MS**

A schematic of the PILS-ESI/MS is shown in Fig. 2. The Particle-into-Liquid Sampler (PILS; Brechtel Manufacturing Inc., Hayward, CA) was developed by Weber et al. (2001) and collects water-soluble aerosol compounds by growing particles into liquid droplets in a supersaturated water environment and then collecting the droplets. A detailed description of the PILS can be found in Sorooshian et al. (2006). The PILS is an established water-soluble aerosol collection technique that has been coupled with various mass analysis methods and was used previously by other laboratories in instrument evaluation studies (e.g. Drewnick et al., 2003; Takegawa et al., 2005; Canagaratna et al., 2007).

The PILS liquid outlet flow was set to 100 µL min$^{-1}$ and was continuously mixed with an acetonitrile flow (100 µL min$^{-1}$). The 1:1 volume mixture of acetonitrile and water was directed toward the custom electrospray ionization source (at ~10 µL min$^{-1}$) of a commercial quadrupole mass spectrometer (Balzers Instruments, QMG 422) operated in negative ion mode for on-line analysis of selected water-soluble organic and inorganic compounds. The

| 1 | electrospray interface involved sample injection at ambient pressure through a fused silica capillary tip (30 µM ID) |
| 2 | with a 2.5 L min$^{-1}$ N$_2$ sheath flow at a spray voltage of -3.5 kV. The MS instrument was modified from the negative- |
| 3 | ion proton-transfer chemical-ionization mass spectrometer (NI-PT-CIMS) described in Veres et al. (2008). The flow |
| 4 | tube was replaced with a stainless steel capillary inlet connected to the front region (I; shown in Fig. 2) held at ~300 |
| 5 | Pa. Ions were focused across this region using a planar DC ion carpet (Anthony et al., 2014) mounted in front of the |
| 6 | orifice leading to the second region (II). The ions were then accelerated through the collisional dissociation chamber |
| 7 | (CDC) and collimated in the octopole ion guide at a total pressure of ~1 Pa (region II). The ions were transferred to |
| 8 | the quadrupole mass spectrometer (region III). The electron multiplier detector was maintained at a pressure of less |
| 9 | than $6.6 \times 10^{-3}$ Pa. |
| 10 | The ESI/MS was calibrated using volumetrically and gravimetrically prepared liquid-phase standards of the |
| 11 | anions associated with the target compounds (e.g. SO$_4^{2-}$, NO$_3^-$, Cl$^-$) (Sigma Aldrich, St. Louis, MO). Anion-specific |
| 12 | calibration factors were calculated from linear least-squares fits of multi-point calibration curves. The uncertainty in |
| 13 | the slope resulted in a maximum uncertainty of ~10% for the compounds tested. The ESI flow rate, solvent |
| 14 | composition, analyte chemical properties, and matrix effects potentially impact the ionization and transmission |
| 15 | efficiencies of compounds (Kostiainen and Kauppila, 2009). For these reasons, experiments were performed under |
| 16 | similar, or as close to identical, conditions as the calibrations for instrument evaluation. The limits of detection for the |
| 17 | anions measured with the PILS-ESI/MS were below ~0.1 µg m$^{-3}$ for the current system and sampling conditions. |
| 18 | Sorooshian et al. (2006) discuss volatility losses in the PILS for several inorganic species and reported negligible loss |
| 19 | with a collection efficiency of ≥96% for mass loadings of Cl$^-$, SO$_4^{2-}$, and NO$_3^-$ ranging from 1-140 µg m$^{-3}$. |
| 20 | Additionally, Orsini et al. (2003) showed the collection efficiency of ≥95% for particles as small as 30 nm diameter |
| 21 | for a 15 L min$^{-1}$ sample flow rate. Ammonium (NH$_4^+$) is the major ion susceptible to volatilization as shown in Ma |
| 22 | (2004), who indicated an underestimation of ~15%. In this study, because we were operating in the negative-ion mode, |
| 23 | we did not measure NH$_4^+$ directly. |

## 2.2 Experimental design

### 2.2.1 Methods for determining gas-phase conversion efficiency

| 26 | The efficiency of conversion of several N-containing gases by the N$_r$ catalyst was determined through |
| 27 | addition of a number of representative compounds that were calibrated independently. The NO signal from the |
| 28 | converted species was then compared to the signal from an NO in N$_2$ standard (5.38 ppmv, Scott-Marrin Inc., |
| 29 | Riverside, CA) that was used as the working standard for this project. Typical calibration levels were in the range of |
| 30 | 50 to 100 ppbv as determined by the mass flow controllers used to mix the standard into the measurement stream. The |
| 31 | standards used for each compound and their associated calibrations were as follows. |
| 32 | The NO$_2$ standard stream was produced from the NO working standard through gas-phase titration with a |
| 33 | small stream of O$_2$ in which O$_3$ had been produced by photolysis at 184.9 nm using a mercury discharge lamp. This |
| 34 | technique is used routinely for NO$_x$ and NO$_y$ measurement systems (Williams et al., 1998) and allows straightforward |
| 35 | determination of NO$_2$ conversion provided care is taken not to over-titrate the NO stream to produce NO$_3$ and therefore |

N$_2$O$_5$. The uncertainty in the NO$_2$ conversion determination is simply the propagated errors in the subtraction of the signals before and after titration.

The ammonia (NH$_3$) conversion was examined using two different NH$_3$ sources, a gas mixture (3.1 ppmv in N$_2$, Scott-Marrin) and a permeation device (Kin-Tek, LaMarque, Texas). Care was taken with these standards to keep them under flow for periods of several days in order to insure any system surfaces were equilibrated. The calibration of these standards was accomplished by ultraviolet (UV) absorption spectroscopy at 184.9 nm wavelength using an instrument described by Neuman et al, (2003), and based on absorption cross-sections reported in the literature (Tannenbaum et al., 1953; Lovejoy, 1999; Froyd, 2002). The uncertainty of NH$_3$ conversion was propagated based on the uncertainties in flow rate and UV absorption determinations.

The hydrogen cyanide (HCN) standard consisted of a commercial gravimetric mixture of HCN in N$_2$ (10 ppmv GASCO Oldsmar, FL), which was mixed into the system using a mass flow controller. The specified uncertainty of this mixture was ±10%, and the standard concentration was verified using long-path Fourier-transform infrared (FTIR) spectroscopy to within the stated uncertainty. The HCN standard was used to produce a gas phase stream of cyanogen chlorine (ClCN) by reaction with chloramine-T, a non-volatile chlorinating agent, which has been described previously (Valentour et al., 1974). To do this, a small stream (5 standard cm$^3$ min$^{-1}$) of the HCN standard was combined with humidified Zero Air (ZA, 60% RH, 30 standard cm$^3$ min$^{-1}$) over a bed packed with glass beads coated with a solution of chloramine-T. The glass beads were prepared by coating glass 3 mm outer diameter (OD) beads with a 2 g 100 mL$^{-1}$ solution and packing ~20 cm$^3$ of them in a 12.7 mm OD PFA tube and flowing ZA over them until dry. The reaction was shown to be essentially 100% (±10%) by proton-transfer reaction mass spectrometry (PTR-MS) when conducted in a humidified atmosphere (RH ≥60%), by FTIR analysis of the HCN and ClCN in the gas stream before and after chlorination.

The isocyanic acid (HNCO) standard was prepared according to the methods described by Roberts et al. (2010), in which the trimer, cyanuric acid, was thermally decomposed at 250°C in a diffusion cell to produce a steady stream of HNCO, which was then calibrated by long-path FTIR spectroscopy. Initially, this source has the potential to produce NH$_3$ as an impurity, most likely because of the presence of trace amounts of water. Keeping the source under flow and above 120°C at all times when not in use was found to reduce the NH$_3$ impurity to negligible levels (<5%), as measured by PTR-MS. The uncertainties in the HNCO standard were propagated from the uncertainties in the HNCO cross section (Northwest-Infrared, PNNL), the NH$_3$ subtraction, and flow rates. Standard streams of both nitrobenzene and trimethylamine were produced using gravimetrically prepared solutions and a commercial liquid calibration device (Ionicon, Innsbruck, Austria). The uncertainties in these liquid calibration standards were estimated from the propagated uncertainties in the solution concentrations and the liquid and gas flow rates.

[revised manuscript text omitted]

 For the liquid concentrations, atomizer conditions, and DMA settings used here, the DMA output size distribution  was a multi-peaked population consisting not only of singly charged particles but also particles with multiple (mostly two or three) charges that.  can contribute significantly to the overall  particle mass.  Hence the particle mass could not be calculated directly from the singly-charged mobility diameter, particle density, and the CPC number concentrations. . We generally used two methods to calculate the particle mass concentrations for these experiments. For the first method, the size distributions were measured using  the scanning mobility particle sizer (SMPS; Wang and Flagan, 1990) function of the DMA (physical diameter, $D_p$ = 1-1000 nm). We used the DMA transfer theory (Knutson and Whitby, 1975; Stolzenburg, 1988) with Wiedensohler's (1988) steady-state charge distribution approximation  to estimate the fraction of multiply charged particles contributing to the CPC number concentration for each diameter setting. There are a number of possible sources of uncertainty using these methods that may include particle losses, DMA transfer function uncertainty, counting uncertainty, and inversion errors. When comparing mass concentrations from the SMPS with those measured by the Nr system, issues with the SMPS-derived size distributions became apparent (discussed separately in Section 3.2.2). For the second method of calculating mass concentrations, we directly measured the diluted, DMA output using the UHSAS. UHSAS particle sizing is a function of the amount of light scattered onto the photodetectors, which.  depends not only on the particle size, but also on the composition-dependent particle refractive index (Bohren and Huffman, 1983; Liu and Daum, 2000; Hand and Kreidenweis 2002; Rosenberg et al., 2012). The UHSAS manufacturer recommended calibration uses PSL microspheres, which are well characterized and have known refractive index (n = 1.58) and shape. Because the UHSAS sizing is sensitive to particle refractive index, a new sizing calibration curve was produced for each studied particle type (i.e. refractive index) using a DMA to select particles for a range of known sizes (Kupc et al., 2018).  These calibration curves were used to retrieve accurate particle size distributions  that properly accounted for the multiply charged particles. .

Differences in particle counting efficiency between the UHSAS and CPC are potentially important. Previous laboratory studies show UHSAS and CPC number concentration comparisons in excellent agreement (Cai et al., 2008;

Kupc et al., 2018), however, occasionally only a ~90% counting efficiency for the UHSAS was observed when compared to the CPC. These differences are attributed to particle coincidence at high concentrations (> 1000 $cm^{-3}$), and to inefficient particle mixing before reaching the instruments. Corrections for particle coincidence were applied (Kupc et al., 2018) though we expect differences due to particle mixing adds an additional 10% uncertainty to the measurements.

The UHSAS and CPC measured particle number concentrations were generally within 10% of each other, however, the CPC values did not require coincidence corrections and had a better signal to noise ratio. Due to problems with measuring SMPS size distributions and requiring coincidence corrections for the UHSAS number concentrations, we used the UHSAS size distributions with the total particle number taken from the CPC measurement to calculate particle mass from total volume and density.

**2.2.2 Nitrogen-containing particles**

[revised manuscript text omitted]

**3.2 Nr particle measurements**

**3.2.1 $N_r$ system set-up and response**

The atomizer output was diluted with particle-free nitrogen and ultra-pure zero air, therefore, the $N_r$ measurement should theoretically be attributed to particles only since no detectable gas-phase nitrogen is added to the sample stream. However, equilibration within the sample lines may result in outgassing and formation of gas-phase compounds affecting total $N_r$ detection. Fig. 3(a) shows the initial response of the $N_r$ system in cleaned inlets for $NaNO_3$. The $N_r$ mass signal tracks the CPC-derived aerosol mass features closely as the aerosol source concentrations fluctuate. Additionally, as different particle sizes are selected by the DMA for $(NH_4)_2SO_4$ (Fig. 3(b)), changes in the total $N_r$ response is fast and precisely tracks the changes in the CPC signal. The potential gas-phase constituents equilibrating in the lines from aerosols in this study include $HNO_3$, HCl, and $NH_3$. If these compounds formed before reaching the $N_r$ catalyst it is likely adsorption and desorption from inlets and tubing surfaces would occur (e.g. Neuman et al., 1999; Yokelson et al., 2003). As an example, the presence of $NH_3$ in Fig. 3(b) (or $HNO_3$ in nitrate containing particles) would be indicated by a delayed and lengthened rise/fall in the $N_r$ response with sudden changes to the input concentrations. However, the total $N_r$ response precisely tracks the CPC signal on rapid timescales (a few seconds) suggesting that gas-phase $NH_3$ was not present in significant quantities. In experiments at exceptionally high aerosol loading of $(NH_4)_2C_2O_4$ (up to several ppmv of total $N_r$ i.e., several thousand µg m$^{-3}$) $N_r$ signal "tailing" was observed suggesting that $NH_3$ was scavenging to the walls of the inlet before the heated quartz tubing.

Marx et al. (2012) reported calculated conversion efficiencies in air sampled from a small chamber for $NaNO_3$, $NH_4NO_3$, and $(NH_4)_2SO_4$ to be 78, 142, and 91%, respectively. The authors suggested the overestimation of $NH_4NO_3$ was a result of its semi-volatile properties under ambient conditions that led to the formation of gaseous $NH_3$ and $HNO_3$ in the chamber. For these reasons, we limit the background artifacts and volatilization effects that may have occurred during chamber filling and sampling in Marx et al. (2012) by sampling immediately following solution atomization through conductive tubing at relatively high sample flow rates. Additionally, we use a DMA to size-select the atomized polydisperse aerosol to evaluate the particle conversion efficiency at several different diameters (100 – 600 nm in 50 nm increments) to investigate the volatilization effects and conversion efficiencies of smaller particles for the extended list of $N_r$-containing aerosols studied in our work.

~~The atomizer output was diluted with particle-free nitrogen and ultra-pure zero air, therefore, the $N_r$ measurement should theoretically be attributed to particles only since no detectable gas-phase nitrogen is added to the sample stream. However, equilibration within the sample lines may result in outgassing and formation of gas-phase compounds affecting total $N_r$ detection. Fig. 3(a) shows the initial response of the $N_r$ system in cleaned inlets for $NaNO_3$. The $N_r$ mass signal tracks the CPC-derived aerosol mass features closely as the aerosol source concentrations~~

**3.1.32.2 Nᵣ particle conversion efficiencyChallenges using the DMA/SMPS to determine Nr-particle conversion**

**efficiency**

[revised manuscript text omitted]

Figure 6 7(a) shows the co-measured $N_r$ and NO concentrations (ppmv). The majority of the $N_r$ system's response is due to the sum of gas-phase $N_r$-constituents that were measured by a Fourier transform infrared spectrometer (FTIR;

Selimovic et al., 2017 2018), an $H_3O^+$ chemical ionization mass spectrometer (Koss et al., 2017 2018), and a broadband cavity enhanced extinction spectrometer (Min et al., 2016) (Fig. 6 7(b)). At the beginning of the burn (pre 10:23 AM)

the average relative percent difference between the total nitrogen signal and the sum of individually measured gas- phase compounds is ~16%, which is less than the combined error of the individual measurements. There is greater disagreement shown in Fig. 6 7(c) (difference is up to ~1 ppmv; up to ~50% relative percent difference) during other stages of the fire. The modified combustion efficiency (MCE) is a measure to estimate the relative contribution of flaming and smoldering combustion that occurred over the course of a fire, where the MCE is defined as the ratio of

$\Delta CO_2 / (\Delta CO_2 + \Delta CO)$ (Yokelson et al., 1996). A higher MCE value (approaching 0.99) designates relatively pure flaming combustion (more complete oxidation) and a lower MCE (~ 0.75–0.84) designates more smoldering combustion. We have shown in our laboratory experiments that there is quantitative $N_r$ particle conversion across the

$N_r$ catalyst, therefore, it is  possible that the residual signals are due to particulate N-containing compounds.

Particulate ammonium may contribute to the excess $N_r$ signal measured during periods dominated by smoldering combustion (MCE < 0.90) . The oxidized N-containing gas phase species are relatively more abundant during the initial part of the fire, so  particulate nitrate  could account 
[revised manuscript text omitted]

Froyd, K. D.: Ion induced nucleation in the atmosphere: Studies of ammonia, sulfuric acid, and water cluster ions, Ph.D., Department of Chemistry, University of Colorado, Boulder, Colorado, 282 pp., 2002.

Fuzzi, S., Baltensperger, U., Carslaw, K., Decesari, S., Denier van der Gon, H., Facchini, M. C., Fowler, D., Koren, I., Langford, B., Lohmann, U., Nemitz, E., Pandis, S., Riipinen, I., Rudich, Y., Schaap, M., Slowik, J. G., Spracklen, D. V., Vignati, E., Wild, M., Williams, M., and Gilardoni, S.: Particulate matter, air quality and climate: lessons learned and future needs, Atmos. Chem. Phys., 15, 8217-8299, doi:10.5194/acp-15-8217-2015, 2015.

Griffith, D. W. T., Mankin, W. G., Coffey, M. T., Ward, D. E., and Riebau, A.: FTIR remote sensing of biomass burning emissions of $CO_2$, CO, $CH_4$, $CH_2O$, NO, $NO_2$, $NH_3$, and $N_2O$, in: Global Biomass Burning: Atmospheric, Climatic, and Biospheric Implications, edited by: Levine, J. S., MIT Press, Cambridge, 230–239, 1991.

Hallquist, M., Wenger, J. C., Baltensperger, U., Rudich, Y., Simpson, D., Claeys, M., Dommen, J., Donahue, N. M., George, C., Goldstein, A. H., Hamilton, J. F., Herrmann, H., Hoffmann, T., Iinuma, Y., Jang, M., Jenkin, M. E., Jimenez, J. L., Kiendler-Scharr, A., Maenhaut, W., McFiggans, G., Mentel, Th. F., Monod, A., Prévôt, A. S. H., Seinfeld, J. H., Surratt, J. D., Szmigielski, R., and Wildt, J.: The formation, properties and impact of secondary organic aerosol: current and emerging issues, Atmos. Chem. Phys., 9, 5155-5236, https://doi.org/10.5194/acp-9-5155-2009, 2009.

[revised manuscript text omitted]

*the number of C atoms in threonine (4).*

[Figure]

**Figure 7.** Timeseries for Fire Sciences Lab 2016 measurements of emissions from a subalpine fir canopy sample (Fire 047). (a) Total reactive nitrogen ($N_r$, red) and nitric oxide (NO, blue) measurements. (b) Comparison of the difference ($N_r$-NO, gold) with the sum of the measured gas phase $N_r$-species (purple). The sum of individually measured gas-phase species in order of abundance include: $NH_3$, HNCO, HCN, HONO, $NO_2$, $CH_3NO_2$, and 40 minor organic nitrogen species. $NO_2$ and HONO were measured by a broadband cavity enhanced extinction spectrometer, HCN and $NH_3$ were measured by FTIR, and all remaining organic species were measured by $H_3O^+$ CIMS. (c) Residual $N_r$ (black) in ppmv with modified combustion efficiency overlaid (MCE, red).

[Figure]

**Figure 8.** The PILS-ESI/MS measured aerosol nitrate mass (blue) and the nitrate measured as $N_r$ (black) ($\mu g \ m^{-3}$) for an atomized solution of $NaNO_3$ (polydisperse). The PILS-ESI/MS trace is shifted to account for the delayed response and the instrument time constant.

[Figure]

**Figure 9.** Scatter plots of PILS-ESI/MS measured versus equivalent anion mass measured as $N_r$ for salts $NaNO_3$ (blue), $NH_4NO_3$ (gold), $(NH_4)_2SO_4$ (red), and $NH_4Cl$ (magenta). The data are 60 s averages and only include times when the atomized aerosol output was relatively constant (i.e. not when concentrations were rising/falling). The slope (1σ) and $R^2$ is shown.

[Figure]

**Figure 10.** The $N_r$ (black) measured, CPC number with UHSAS size (blue) calculated, UHSAS number and size (red) calculated, and PILS-ESI/MS(green) measured aerosol concentrations ($\mu g\ m^{-3}$) for anions of DMA size selected aerosol for salts of (a) $NaNO_3$, (b) $(NH_4)_2SO_4$, (c) $NH_4Cl$, and (d) $(NH_4)_2C_2O_4$. The PILS-ESI/MS traces were shifted in time several minutes early to account for the delayed instrument response time.

**Tables**

**Table 1.** Conversion efficiencies of $N_r$ compounds by the Pt/MoOx catalyst system

| Compound | Conversion efficiency (%) | Calibration method | Reference |
|---|---|---|---|
| Nitrogen Dioxide, $NO_2$ | 99 ± 2 | Titration of NO standard by $O_3$ | Williams et al., 1998 |
| Ammonia, $NH_3$ | 105-110 ± 15 | Permeation tube or gas mixture, UV absorbance at 184.9nm | Neuman et al., 2003 |
| Hydrogen cyanide, HCN | 101-102 ± 10 | Gravimetric gas mixture | GASCO, Oldsmar, FL. |
| Cyanogen chloride, ClCN | 98 ± 10 | Conversion of HCN standard with Chloramine-T | Valentour et al., 1974 |
| Isocyanic Acid, HNCO | 100 ± 25 | Decomposition of the trimer, FTIR | Roberts et al., 2010 |
| Nitrobenzene, $C_6H_5NO_2$ | 95 ± 15 | Liquid calibration unit, liquid flow and gravimetric concentration | Ionicon, Innsbruck, Austria |
| Triethyl amine, $(C_2H_5)_3N$ | 95 ± 15 | Liquid calibration unit, liquid flow and gravimetric concentration | Ionicon, Innsbruck, Austria |

**Table 2.** Particle conversion efficiencies (%) with uncertainties (one standard deviation) in parentheses. The sizing accuracy is ~± 2.5% using NIST-traceable PSLs for 150 –500 nm spheres as our calibration standard.

| Diameter (nm) | $NaNO_3$ | $(NH_4)_2SO_4$ | $NH_4Cl$ | $(NH_4)_2C_2O_4$ |
|---|---|---|---|---|
| 100 | 88.4(18.3) | 100.6(3.0) | 89.2(5.9) | 91.0(3.5) |
| 150 | 94.0(10.9) | 96.5(2.5) | 93.4(4.7) | 89.0(6.6) |
| 200 | 98.6(4.0) | 98.8(4.8) | 93.6(4.2) | 90.2(5.1) |
| 250 | 101(3) | 100(3) | 98.3(3.7) | 94.7(5.6) |
| 300 | 104(6) | 102(9) | 101(3) | 97.0(6.2) |
| 350 | 102(6) | 101(9) | 98.5(5.2) | 101(13) |
| 400 | 103(8) | 100(8) | 100(6) | 94.7(7.4) |
| 450 | 95.1(4.5) | 110(4) | 103(6) | - |
| 500 | 103(15) | 109(17) | 124(11) | 96.3(7.6) |
| 600 | 83.2(8.7) | 91.9(5.5) | - | 82.5(8.4) |
| Average | 97.3(7.1) | 101(5) | 100(10) | 92.9(5.4) |